# Tracing the development and lifespan change of population-level structural asymmetry in the cerebral cortex

**James M Roe[1]\***, **Didac Vidal-Pineiro[1]**, **Inge K Amlien[1]**, **Mengyu Pan[1]**, **Markus H Sneve[1]**, **Michel Thiebaut de Schotten[2,3]**, **Patrick Friedrich[4]**, **Zhiqiang Sha[5]**, **Clyde Francks[5,6,7]**, **Espen M Eilertsen[8]**, **Yunpeng Wang[1]**, **Kristine B Walhovd[1,9]**, **Anders M Fjell[1,9]**, **René Westerhausen[10]**

[1]Center for Lifespan Changes in Brain and Cognition (LCBC), Department of Psychology, University of Oslo, Oslo, Norway; [2]Groupe d'Imagerie Neurofonctionnelle, Institut des Maladies Neurodégénératives-UMR 5293, CNRS, CEA, University of Bordeaux, Bordeaux, France; [3]Brian Connectivity and Behaviour Laboratory, Sorbonne University, Paris, France; [4]Institute of Neuroscience and Medicine, Research Centre Jülich, Jülich, Germany; [5]Language and Genetics Department, Max Planck Institute for Psycholinguistics, Nijmegen, Netherlands; [6]Donders Institute for Brain, Cognition and Behaviour, Radboud University, Nijmegen, Netherlands; [7]Department of Human Genetics, Radboud University Medical Center, Nijmegen, Netherlands; [8]PROMENTA Research Center, Department of Psychology, University of Oslo, Oslo, Norway; [9]Department of Radiology and Nuclear Medicine, Oslo University Hospital, Oslo, Norway; [10]Section for Cognitive and Clinical Neuroscience, Department of Psychology, University of Oslo, Oslo, Norway

**\*For correspondence:**
j.m.roe@psykologi.uio.no

**Competing interest:** The authors declare that no competing interests exist.

**Abstract** Cortical asymmetry is a ubiquitous feature of brain organization that is subtly altered in some neurodevelopmental disorders, yet we lack knowledge of how its development proceeds across life in health. Achieving consensus on the precise cortical asymmetries in humans is necessary to uncover the developmental timing of asymmetry and the extent to which it arises through genetic and later influences in childhood. Here, we delineate population-level asymmetry in cortical thickness and surface area vertex-wise in seven datasets and chart asymmetry trajectories longitudinally across life (4–89 years; observations = 3937; 70% longitudinal). We find replicable asymmetry interrelationships, heritability maps, and test asymmetry associations in large–scale data. Cortical asymmetry was robust across datasets. Whereas areal asymmetry is predominantly stable across life, thickness asymmetry grows in childhood and peaks in early adulthood. Areal asymmetry is low-moderately heritable (max $h^2_{SNP}$ ~19%) and correlates phenotypically and genetically in specific regions, indicating coordinated development of asymmetries partly through genes. In contrast, thickness asymmetry is globally interrelated across the cortex in a pattern suggesting highly left-lateralized individuals tend towards left-lateralization also in population-level right-asymmetric regions (and vice versa), and exhibits low or absent heritability. We find less areal asymmetry in the most consistently lateralized region in humans associates with subtly lower cognitive ability, and confirm small handedness and sex effects. Results suggest areal asymmetry is developmentally stable and arises early in life through genetic but mainly subject-specific stochastic effects, whereas childhood developmental growth shapes thickness asymmetry and may lead to directional variability of global thickness lateralization in the population.

## Editor's evaluation

Roe et al. provide a large-sample analysis of hemispheric lateralisation in brain structure, synthesising local cortical thickness and surface area data from 7 different datasets. The study provides a rich descriptive catalogue of phenomena related to hemispheric anatomical asymmetries. These results are convincing and will prove an important point of reference to neuroscientists who might want to compare their own future results to the ones from this large and varied data set.

## Introduction

The brain's hemispheres exhibit high contralateral symmetry (*van Kesteren and Kievit, 2021*; *Stark et al., 2008*), homotopic regions are under similar genetic influence (*Schmitt et al., 2018*; *Chen et al., 2013*; *Eyler et al., 2014*) and show highly correlated developmental change (*Schmitt et al., 2018*; *Raznahan et al., 2011*). Despite this, structural asymmetry is also a ubiquitous aspect of brain organization (*Kong et al., 2018*; *Chiarello et al., 2016*). Cortical thickness and surface area are known to exhibit distinct asymmetry patterns (*Kong et al., 2018*; *Meyer et al., 2014*), albeit reported inconsistently (*Kong et al., 2018*; *Chiarello et al., 2016*; *Roe et al., 2021*; *Li et al., 2015*; *Luders et al., 2006*; *Lyttelton et al., 2009*; *Sha et al., 2021a*; *Zhou et al., 2013*; *Maingault et al., 2016*; *Shaw et al., 2009*; *Lou et al., 2020*; *Hamilton et al., 2007*; *Zhou et al., 2018*; *Koelkebeck et al., 2014*; *Plessen et al., 2014*). Yet achieving consensus on cortical asymmetries in humans is a prerequisite to uncover the genetic-developmental and lifespan influences that shape and alter them. Although an extensive literature in search of structural asymmetry deviations in various conditions and disorders is in several cases being challenged by newer data (*Kong et al., 2022*), at least some aspects of cortical asymmetry are confirmed to be subtly reduced in neurodevelopmental disorders such as autism (*Postema et al., 2019*; *Sha et al., 2022*), but also through later life influences such as aging, and Alzheimer's disease (*Roe et al., 2021*; *Thompson et al., 2007*). Hence, altered cortical asymmetry at various lifespan stages may be associated with reduced brain health. However, it is currently unknown how cortical asymmetry development proceeds across life in health, because no previous study has charted cortical asymmetry trajectories from childhood to old age using longitudinal data.

Compounding the lack of longitudinal investigation, previous large-scale studies do not delineate the precise brain regions exhibiting robust cortical asymmetry, relying on brain atlases with predefined anatomical boundaries that may not conform well to the underlying asymmetry of cortex (*Kong et al., 2018*; *Sha et al., 2021b*). Taking an atlas-free approach to delineate asymmetries that reliably reproduce across international samples as starting point (i.e. population-level asymmetries) would better enable mapping of the developmental principles underlying structural cortical asymmetries, as well as the genetic and individual-specific factors associated with cortical lateralization. Furthermore, such an approach would help resolve the many reported inconsistencies for cortical asymmetry maps – for example reports of both right- (*Maingault et al., 2016*; *Lou et al., 2020*; *Hamilton et al., 2007*; *Williams et al., 2022*) and left- (*Roe et al., 2021*; *Li et al., 2015*; *Luders et al., 2006*; *Plessen et al., 2014*) thickness lateralization in medial and lateral prefrontal cortex (PFC; *Chiarello et al., 2016*; *Zhou et al., 2013*; *Maingault et al., 2016*; *Shaw et al., 2009*; *Koelkebeck et al., 2014* and *Roe et al., 2021*; *Luders et al., 2006*; *Sha et al., 2021a*; *Plessen et al., 2014*; *Williams et al., 2023*), and of right- (*Maingault et al., 2016*; *Li et al., 2014*) and left areal lateralization of superior temporal sulcus (STS) (*Kong et al., 2018*; *Bain et al., 2019*; *Remer et al., 2017*) – while serving as a high-fidelity phenotype for future brain asymmetry studies to complement existing low-resolution atlases (*Kong et al., 2018*).

Determining the developmental and lifespan trajectories of cortical asymmetry may shed light on how cortical asymmetries are shaped through childhood or set from early life, and provide evidence of the timing of expected brain change in normal development. Although important in and of itself, this would also provide a useful normative reference, as subtly altered cortical asymmetry – in terms of both area and thickness – has been linked at the group-level to neurodevelopmental disorders along the autism spectrum, suggesting altered lateralized neurodevelopment may be a relevant outcome in at least some cases of developmental perturbation (*Postema et al., 2019*; *Sha et al., 2022*). For areal asymmetry, surprisingly few studies have charted developmental (*Li et al., 2014*; *Remer et al., 2017*) or aging-related effects (*Kong et al., 2018*; *Williams et al., 2022*), although indirect evidence in neonates suggests adult-like patterns of areal asymmetry are evident at birth (*Li et al., 2014*;

*Williams et al., 2023*) and may exhibit little change from birth to 2 years (*Li et al., 2014*) despite rapid and concurrent developmental cortical expansion (*Li et al., 2013*). For thickness asymmetry, longitudinal increases in asymmetry have been shown during the first two years of life (*Li et al., 2015*), with suggestions of rapid asymmetry growth from birth to 1 year (*Li et al., 2015*), and potentially continued growth until adolescence (*Nie et al., 2013*). However, previous lifespan studies mapped thickness asymmetry linearly across cross-sectional developmental and adult age-ranges (*Zhou et al., 2013*; *Plessen et al., 2014*), mostly concluding thickness asymmetry is minimal in infancy and maximal age ~60. In contrast, recent work established thickness asymmetry shows a non-linear decline from 20 to 90 years that is reproducible across longitudinal aging cohorts (*Roe et al., 2021*). Thus, although offering viable developmental insights (*Zhou et al., 2013*; *Plessen et al., 2014*), previous lifespan studies of thickness asymmetry do not accurately capture the aging process, and likely conflate non-linear developmental and aging trajectories with linear models. A longitudinal exploration of the lifespan trajectories of thickness asymmetry accounting for dynamic change is needed to further knowledge of normal human brain development.

Correlations between cortical asymmetries may provide a window on asymmetries formed under common genetic or developmental influences. Contemporary research suggests brain asymmetries are complex, multifactorial and independent (i.e. uncorrelated) traits (*Rentería, 2012*; *Francks, 2015*; *Neubauer et al., 2020*), contrasting earlier theories emphasizing a single (*Annett, 1998*; *Annett, 1964*) or predominating factor controlling various cerebral lateralizations (*McManus and Bryden, 1991*; *Geschwind and Galaburda, 1985*). Yet while there has been much research on whether asymmetries of various morphometric measures (*Chiarello et al., 2016*; *Maingault et al., 2016*; *Koelkebeck et al., 2014*) or imaging modalities relate to one another (*Bain et al., 2019*), few have focused on interregional relationships between asymmetries derived from the same metric. Where reported, evidence suggests cortical asymmetries are mostly independent (*Sha et al., 2021b*; *Guadalupe et al., 2015*; *Chiarello et al., 2013*) – in line with a multifactorial view (*Bain et al., 2019*; *Rentería, 2012*; *Francks, 2015*; *Liu et al., 2009*). Currently, it is unknown whether or which cortical asymmetries are reliably correlated within individuals, though this may signify coordinated development of left-right brain asymmetries through genetic or lifespan influences – a genetic or later developmental account depending on trait heritability and whether phenotypic correlations are underpinned by genetic correlations.

Finally, altered development of cerebral lateralization has been widely hypothesized to relate to average poorer cognitive outcomes (*Plessen et al., 2014*; *Crow et al., 1998*; *Hirnstein et al., 2010*). Specifically in the context of cortical asymmetry, however, although one previous study reported larger thickness asymmetry may relate to better verbal and visuospatial cognition (*Plessen et al., 2014*), phenotypic asymmetry-cognition associations have been rarely reported (*Plessen et al., 2014*; *Moodie et al., 2020*; *Yeo et al., 2016*), conflicting (*Moodie et al., 2020*; *Yeo et al., 2016*), not comparable (*Plessen et al., 2014*; *Moodie et al., 2020*), and to date remain untested in large-scale data. Still, recent work points to small but significant overlap between genes underlying multivariate brain asymmetries and those influencing educational attainment and specific developmental disorders impacting cognition (*Sha et al., 2021b*), indicating either pleiotropy between non-related traits or capturing shared genetic susceptibility to altered brain lateralization and cognitive outcomes. However, most large-scale studies examining factors widely assumed important for asymmetry have used brain atlases with limited spatial precision (*Kong et al., 2018*; *Kong et al., 2022*; *Sha et al., 2021b*). Accordingly, such studies did not detect associations with handedness (*Kong et al., 2018*; *Wiberg et al., 2019*) that were not found until a recent study applied higher resolution (i.e. vertex-wise) mapping in big data (*Sha et al., 2021a*). Therefore, as a final step, we reasoned that combining an optimal delineation of population-level cortical asymmetries with big data would optimize detection and quantification of the effects of factors purportedly related to asymmetry, namely cognitive ability, handedness, and sex.

Here, we first aimed to delineate population-level cortical areal and thickness asymmetries using vertex-wise analyses and their overlap in seven international datasets. With a view to gaining insight into cortical asymmetry development, we then aimed to trace a series of lifespan and genetic analyses. Specifically, we chart the developmental and lifespan trajectories of cortical asymmetry for the first time longitudinally across the lifespan. Next, we examine phenotypic interregional asymmetry correlations, under the assumption correlations indicate coordinated development of left-right asymmetries

through genes or lifespan influences. To shed light on the extent to which differences in asymmetry are genetic, we test heritability of asymmetry using genome-wide single nucleotide polymorphism (SNP) and extended twin data, and examine whether or not phenotypic correlations are underpinned by genetic correlations suggestive of coordinated development through genes. Finally, we screen our set of robust, population-level asymmetries for association with general cognitive ability and factors purportedly related to asymmetry in UK Biobank (UKB) (*Miller et al., 2016*). Based on findings of aging-related dedifferentiation in thickness asymmetry (*Roe et al., 2021*), we hypothesized trajectories of cortical thickness would show developmental growth in thickness asymmetry (i.e. differentiation), but remained agnostic regarding lifespan areal asymmetry development.

## Results

### Population-level asymmetry of the cerebral cortex

First, to delineate cortical regions exhibiting population-level areal and thickness asymmetry, we assessed asymmetry vertex-wise in 7 independent adult samples and quantified overlapping effects (Methods). Areal asymmetries were highly consistent across all 7 datasets (*Figure 1A*): the spatial correlation between surface AI maps ranged from *r*=0.88 to 0.97 (*Figure 1C*). Across all seven datasets (*Figure 1D*), overlapping effects for strong leftward areal asymmetry were observed in a large cluster in supramarginal gyrus (SMG) that spanned postcentral gyrus, planum temporale and primary auditory regions (and conformed well to their anatomical boundaries; see *Figure 1—figure supplement 1A* for significance), anterior insula and temporal cortex, rostral anterior cingulate, medial superior frontal cortex, and precuneus, the latter spanning parahippocampal and entorhinal cortex. Overlapping effects for strong rightward areal asymmetry were evident in cingulate, inferior parietal cortex, STS, medial occipital cortex, medial PFC (mPFC) and rostral middle frontal cortex (*Figure 1D*). This global pattern agrees with previous reports (*Kong et al., 2018*; *Lyttelton et al., 2009*; *Sha et al., 2021a*).

For thickness, an anterior-posterior pattern of left-right asymmetry was evident in most datasets (*Figure 1B*), consistent with more recent reports (*Kong et al., 2018*; *Roe et al., 2021*; *Sha et al., 2021a*; *Plessen et al., 2014*; *Williams et al., 2022*). Though spatial correlations between AI maps from independent datasets were high, they were notably more variable (*r*=0.33 - 0.93; *Figure 1C*); HCP showed lower correlation with all datasets (*r*=0.33 - 0.46) whereas all other datasets correlated highly (min *r*=0.78). Consistent leftward thickness asymmetry effects were evident in cingulate, post-central gyrus, and superior frontal cortex (*Figure 1E*), whereas consistent effects for rightward thickness asymmetry were evident in a large cluster in and around STS (*Figure 1E*; *Figure 1—figure supplement 1B*), insula, lingual gyrus, parahippocampal and entorhinal cortex. Of note, both areal and thickness asymmetry extended beyond these described overlapping effects (*Figure 1—figure supplements 1 and 2* & *Figure 1—figure supplement 5*).

Based on effect size criteria (*Figure 1D–E*; Methods), we derived a set of robust clusters exhibiting population-level areal (14 clusters) and thickness asymmetry (20 clusters) for further analyses (see *Supplementary file 1E-F* for anatomical descriptions). The proportion of individuals lateralized in the population direction in each cluster was highly similar across datasets, on average ranging between 61% and 94% for area, and 57–90% for thickness (*Figure 1F–I*). We then formally compared our approach to asymmetry estimates derived from a gyral-based atlas often used to assess asymmetry (*Kong et al., 2018*; *Koelkebeck et al., 2014*; *Sha et al., 2021b*), finding fairly poor correspondence with the vertex-wise structure of cortical asymmetry for atlas-based regions, particularly for thickness asymmetry (*Figure 1—figure supplement 3*).

### Lifespan trajectories of population-level cortical asymmetries

Having delineated regions exhibiting population-level areal and thickness asymmetry, we aimed to characterize the developmental and lifespan trajectories of cortical asymmetry from early childhood to old age, using a lifespan sample incorporating dense longitudinal data (Methods). For this, we used the mixed-effects LCBC lifespan sample covering the full age-range (4–89 years). To account for non-linear lifespan change, we used Generalized Additive Mixed Models (GAMMs) to model the smooth left- (LH) and right hemisphere (RH) age-trajectories in our robust clusters (Methods).

In all clusters, the homotopic areal trajectories revealed areal asymmetry was strongly established already by age ~4, and the lifespan trajectories of both leftward (*Figure 2A*) and rightward (*Figure 2B*)

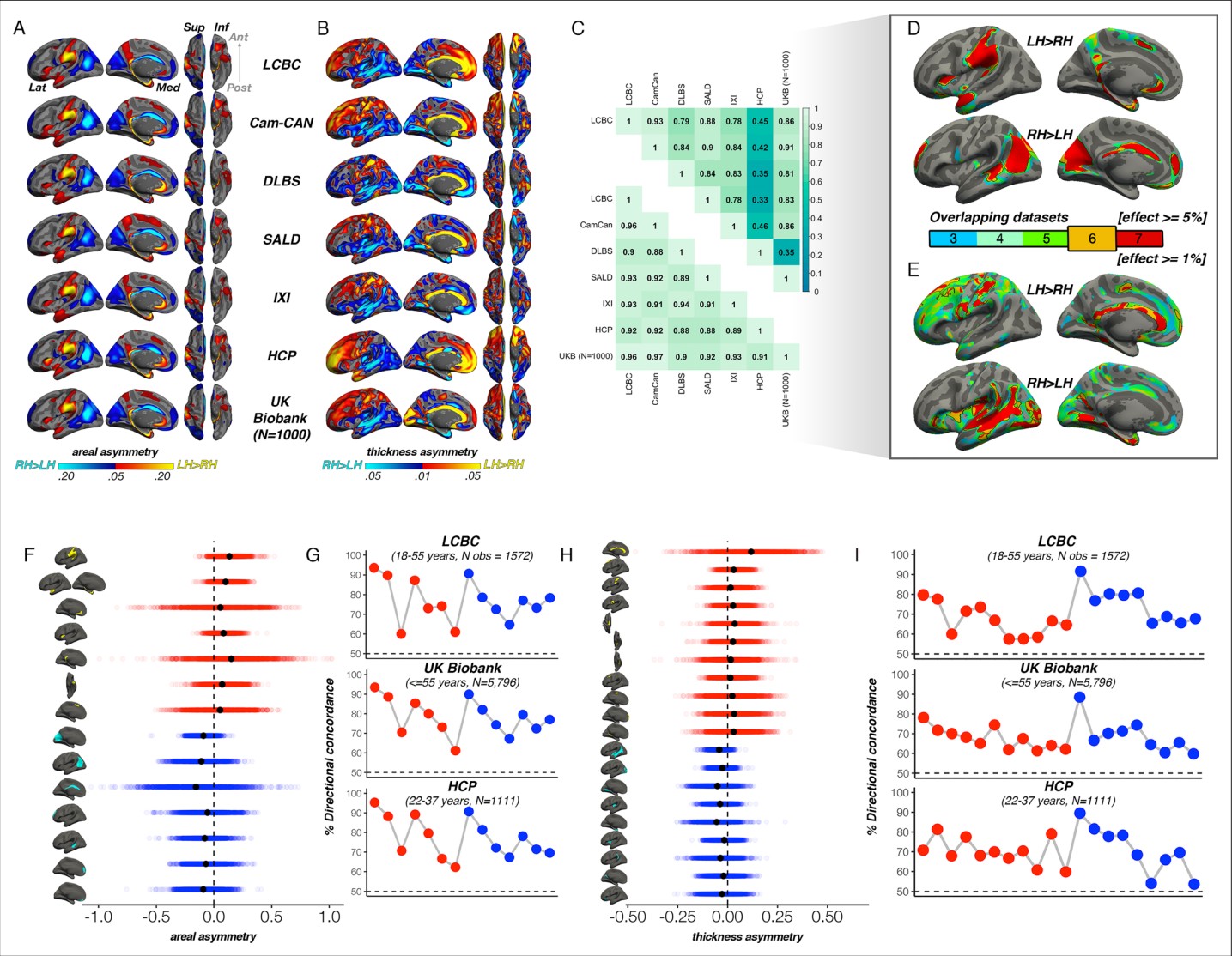

**Figure 1.** Population-level asymmetry of the cerebral cortex. (**A**) Mean areal and (**B**) thickness asymmetry in each dataset. Warm and cold colours depict leftward and rightward asymmetry, respectively. (**C**) Spatial overlap (Pearson's r) of the unthresholded maps between datasets for areal (lower matrix) and thickness asymmetry (upper). (**D**) Overlapping effects across datasets were used to delineate clusters exhibiting population-level areal (lower threshold = 5%) and (**E**) thickness asymmetry (lower threshold = 1%) based on a minimum 6-dataset overlap (black outlined clusters). (**F, H**) Raw distribution of the individual-level asymmetry index (AI) in adults extracted from clusters exhibiting areal and thickness asymmetry, respectively. Mean AI's are in black, Raw distributions are shown for the LCBC (18–55 years) dataset with mixed effects data (cluster-wise outliers defined in lifespan analysis removed on a region-wise basis; Methods; *Supplementary file 1E-F*). X-axis denotes the AI of the average thickness and area of a vertex within the cluster. (**G, I**) Proportion of individuals with the expected directionality of asymmetry within each cluster exhibiting areal and thickness asymmetry, respectively, shown for the three largest adult datasets. The X-axes in G and I are ordered according to the clusters shown in F and H, respectively. Lat = lateral; Med = medial; Post = posterior; Ant = anterior; Sup = superior; Inf = inferior.

The online version of this article includes the following figure supplement(s) for figure 1:

**Figure supplement 1.** Significance of asymmetry effects across samples.

**Figure supplement 2.** Unthresholded maps.

**Figure supplement 3.** Comparison of vertex-wise and atlas-based asymmetry estimates.

**Figure supplement 4.** HCP pipeline.

**Figure supplement 5.** Unthresholded asymmetry effects analyzed using a standard brain atlas with no cross-hemispheric registration.

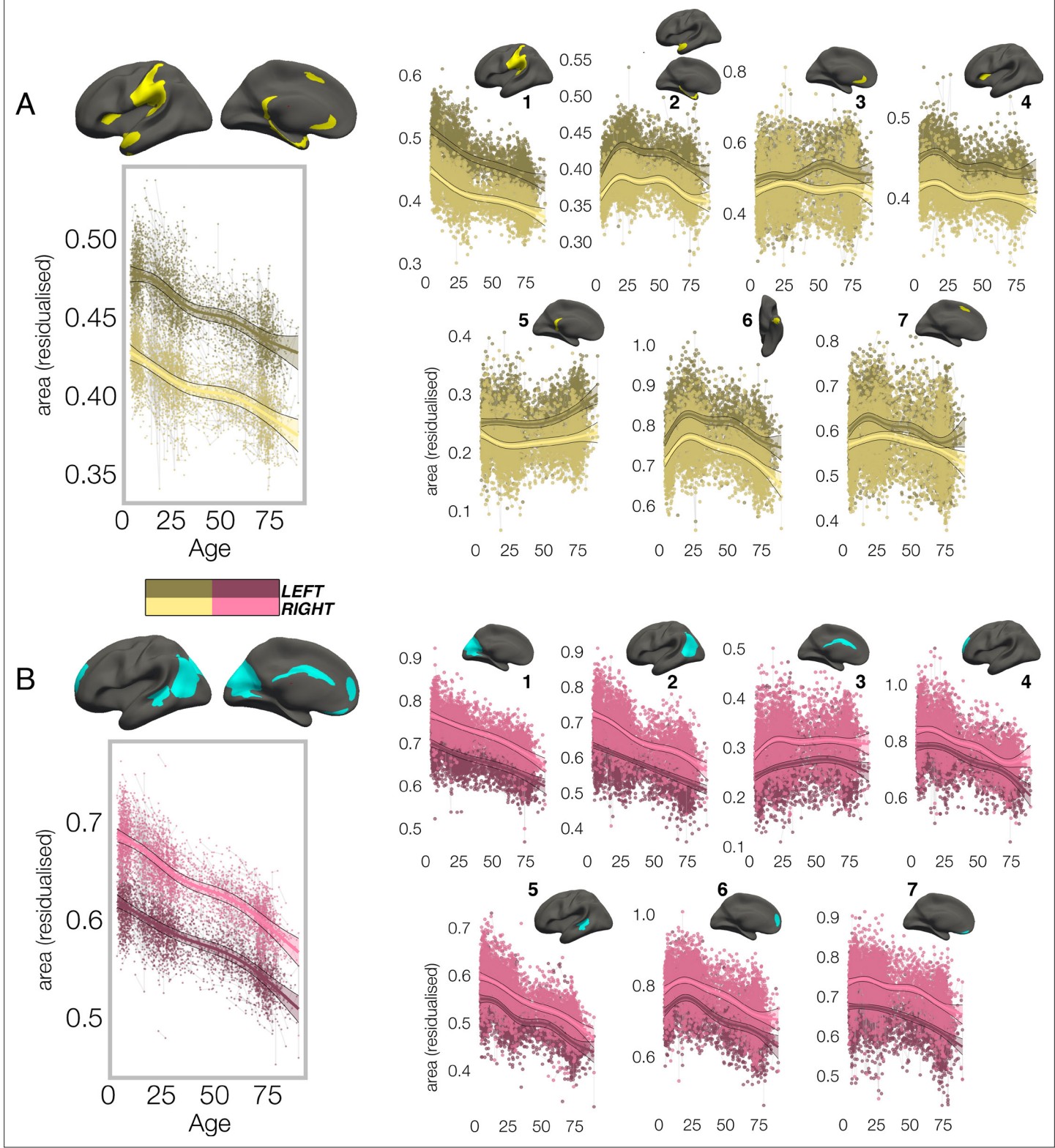

**Figure 2.** Lifespan trajectories of population-level areal asymmetry. Homotopic lifespan trajectories of surface area in clusters exhibiting population-level (**A**) leftward (yellow plots; yellow clusters) and (**B**) rightward (pink plots; blue clusters) areal asymmetry (mm²). Larger plots on the left show the mean age trajectory across all clusters exhibiting leftward (top) and rightward (bottom) asymmetry. Note that the unit of measurement is the average surface area of a vertex within the cluster. Dark colours correspond to LH trajectories. All age trajectories were fitted using GAMMs. Data is residualized for sex, scanner and random subject intercepts. Clusters are numbered for reference. As outliers were removed on a region-wise basis (Methods), the number of observations underlying the plots range from 7862 to 7874 (see *Supplementary file 1E*).

asymmetries were largely parallel. Specifically, a large left-asymmetric region in and around SMG/peri-sylvian (#1; *Figure 2A*) showed strong asymmetry by age ~4 that was largely maintained throughout life through steady aging-associated decline of both hemispheres, whereas leftward asymmetry of temporal cortex (#2,6) and anterior insular (#4) was maintained through developmental expansion and aging-associated decline of both hemispheres. Others (retrosplenial #5; mPFC #3,7) showed growth from pre-established asymmetry and more variable lifespan trajectories. On the other side, rightward asymmetries showed largely preserved asymmetry through aging-associated decline of both hemi-spheres (*Figure 2B*; medial occipital #1; lateral parietal #2; STS #5; orbitofrontal #7), through bilateral developmental expansion and aging-associated decline (mPFC #6), or steadily expanding bilateral surface area until mid-life (cingulate; #3). There was also little indication of relative hemispheric differ-ences during cortical developmental expansion from 4 to 30 years (*Figure 3—figure supplement 3A*) or aging from 30 to 89 years (*Figure 3—figure supplement 5*). Though lifespan areal asymmetry trajectories did show significant change at some point throughout life in most clusters (*Supplementary file 1E*), factor-smooth GAMM interaction analyses confirmed that areal asymmetry was signifi-cantly different from zero across the entire lifespan in all clusters (*Figure 3—figure supplements 1–2*), and the average trajectories across all leftward and rightward clusters were clearly parallel (although still both exhibited a significant difference; bordered plots in *Figure 2A–B*; *Supplementary file 1E*).

In contrast, though homotopic trajectories of thickness clusters were more variable, they were non-parallel, and mostly characterized by developmental increase in thickness asymmetry from age 4–30, through seemingly unequal rates of continuous thinning between hemispheres (*Figure 3*; *Figure 3—figure supplements 1–3*). Importantly, in 10/20 clusters the data indicated developmental increase in thickness asymmetry corresponded to a significant relative hemispheric difference in the rate of developmental thinning. Mostly, these conformed to a pattern whereby the thicker homotopic hemi-sphere thinned comparatively slower (*Figure 4*): leftward thickness asymmetry developed through comparatively slower thinning of the LH that was significant in 6 clusters (superior, lateral and medial PFC #2 #8 #10, precentral #4, inferior temporal #6, calcarine #11; *Figure 3A*; *Figure 4*), whereas rightward asymmetry developed through significantly slower RH thinning (STS #1, planum temporale #7, anterior insula #9; *Figure 3A*; *Figure 4*), or significantly faster RH thickening (#5 entorhinal). Only one other cluster exhibited a relative hemispheric difference seemingly driven by faster thinning of the thicker hemisphere (#8 posterior cingulate; *Figure 4*). In these clusters, asymmetry development was generally evident until a peak in early adulthood (median age at peak = 24.3; see *Figure 4*) for both leftward and rightward clusters, around a point of inflection to less developmental thinning (see also *Figure 3—figure supplement 4*). The average trajectories across all leftward and rightward clusters also indicated developmental asymmetry increase (bordered plots; *Figure 3*). Despite the developmental growth, factor-smooth GAMMs nevertheless confirmed the developmental founda-tion for thickness asymmetry was already established by age ~4 (95% of clusters exhibited small but significant asymmetry at age ~4; *Figure 3—figure supplement 2B*), and again asymmetry trajectories showed significant change at some point throughout life (*Supplementary file 1F*). Across clusters delineated here we observed little evidence aging-related change from 30 to 89 years corresponded to a relative hemispheric difference in the rate of aging-related thinning, except in regions overlap-ping with our previous report (*Roe et al., 2021*; e.g. mPFC #10; *Figure 3—figure supplement 5B*; *Figure 3—figure supplement 4*). Thus, across population-level thickness asymmetries, the data indi-cated either developmental growth in asymmetry, or conserved relative asymmetry through develop-ment and aging despite absolute asymmetry change. Results were robust to varying the number of knots used to estimate trajectories (*Figure 3—figure supplement 1*).

## Interregional asymmetry correlations

We then investigated which cortical asymmetries correlate within individuals (AI's corrected for age, sex, scanner; Methods). For areal asymmetry, a common covariance structure between asymme-tries was detectable across datasets: Mantel tests revealed the correlation matrices derived inde-pendently in LCBC, UKB and HCP data all correlated almost perfectly ($r \geq 0.97$, all $p<9.9e^{-5}$; *Figure 5A*; *Figure 5—figure supplement 1A*). The highest correlations (or 'hotspots') all reflected positive correlations between regions that are on average left-asymmetric and regions that are on average right-asymmetric (i.e. higher leftward asymmetry in one region related to higher rightward asymmetry in another; *Figure 5A* black outline); leftward asymmetry in SMG/perisylvian (#1 L) was related to

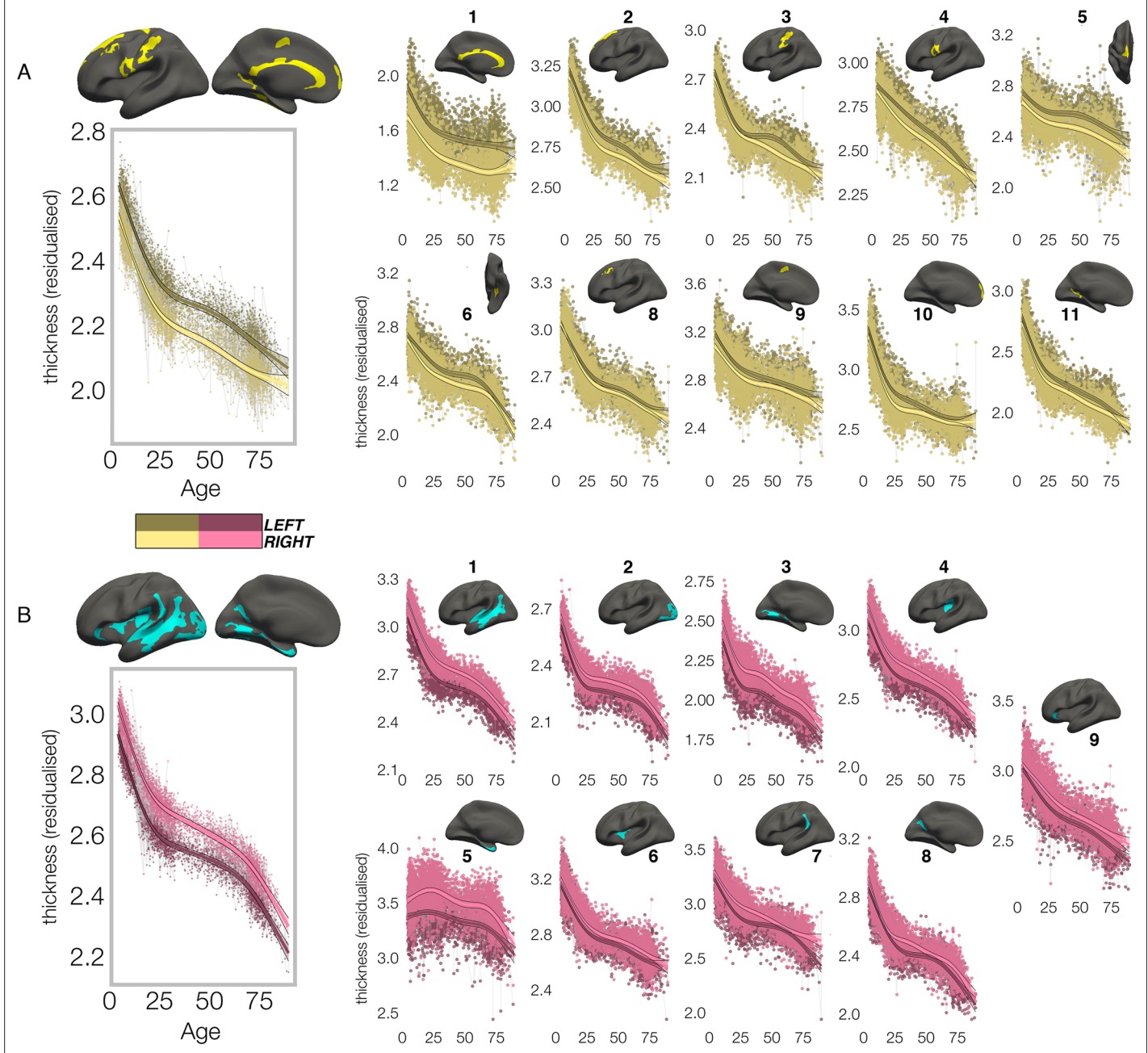

**Figure 3.** Lifespan trajectories of population-level thickness asymmetry. Homotopic lifespan trajectories of cortical thickness in clusters exhibiting population-level (**A**) leftward (yellow plots; yellow clusters) and (**B**) rightward (pink plots; blue clusters) thickness asymmetry (mm). Larger plots on the left show the mean age trajectory across all clusters exhibiting leftward (top) and rightward (bottom) asymmetry. Dark colours correspond to LH trajectories. All age trajectories were fitted using GAMMs. Data is residualized for sex, scanner, and random subject intercepts. Clusters are numbered for reference. As outliers were removed on a region-wise basis (Methods), the number of observations underlying the plots range from 7856 to 7874 (see *Supplementary file 1F*).

The online version of this article includes the following figure supplement(s) for figure 3:

**Figure supplement 1.** Knot comparison.

**Figure supplement 2.** Smooth Age x Hemisphere interactions.

**Figure supplement 3.** Relative developmental trajectories.

**Figure supplement 4.** Lifespan thickness trajectories in regions exhibiting age-related change in asymmetry.

**Figure supplement 5.** Relative aging trajectories.

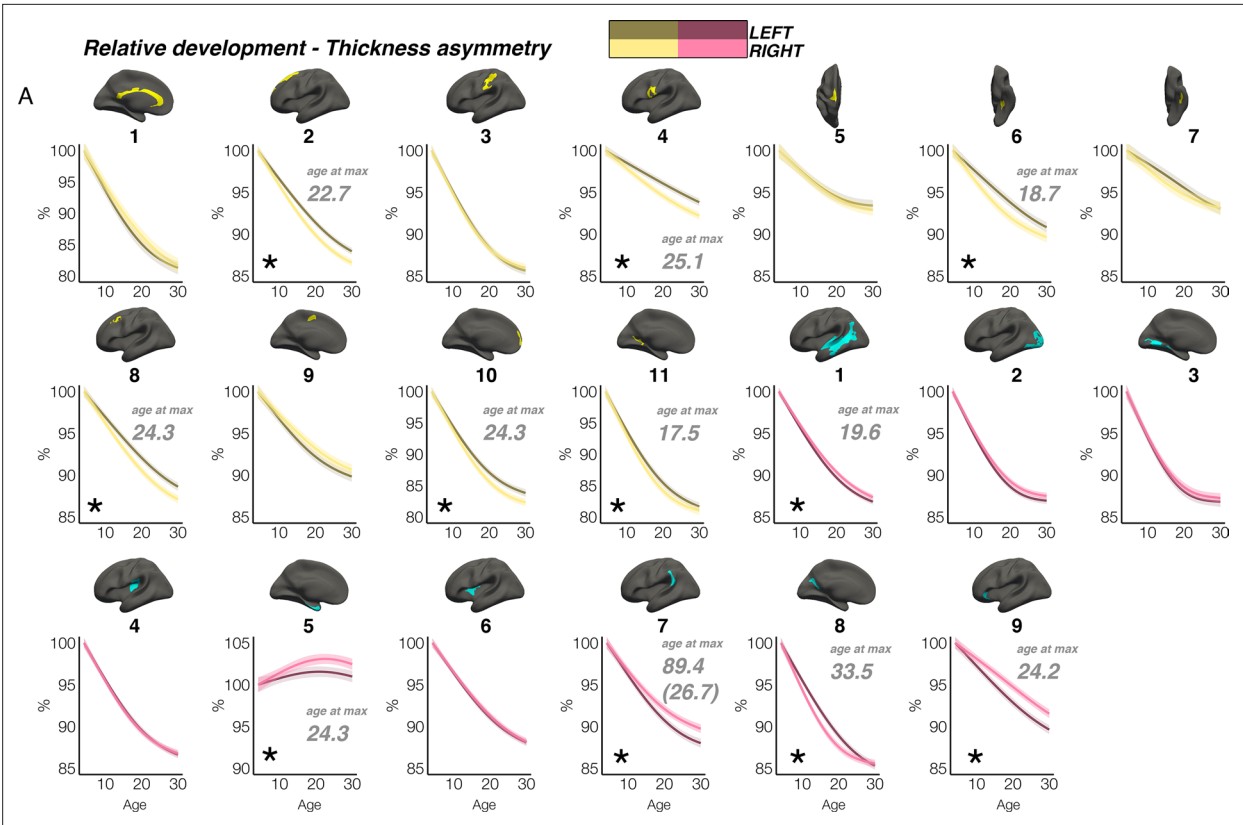

**Figure 4.** Relative developmental trajectories of homotopic cortical thickness in clusters exhibiting population-level thickness asymmetry. To highlight development, the x-axis covers the age-range 4–30 years, although relative change was calculated from full lifespan models (Methods). Darker trajectories indicate LH trajectories. Shaded areas indicate 95% CI. A relative difference in developmental thinning rates between hemispheres is suggested if the CI's of relative LH and RH trajectories diverge to be non-overlapping (denoted with *). These indications should be interpreted in combination with the full lifespan GAMM trajectories shown in *Figure 3*. Where relative hemispheric differences are indicated, age at the point of maximum thickness asymmetry across life is denoted in grey (Methods). Since one cluster (**R7**) was estimated to exhibit maximum thickness asymmetry at age 89.4 (see *Figure 3*), age at maximum asymmetry for the developmental peak is also given in parentheses (*Figure 3*). As outliers were removed on a region-wise basis (Methods), the number of observations underlying the plots range from 7856 to 7874 (see *Supplementary file 1F*).

higher rightward asymmetry in inferior parietal cortex (#2 R; r=0.48 [LCBC]), leftward anterior cingulate asymmetry (ACC; #3 L) was related to higher rightward asymmetry in mPFC (#6 R, r=0.47), and leftward asymmetry in a superior frontal cluster (#7 L) was related to rightward asymmetry in the cingulate (#3 R, r=0.68). None of the relationships could be explained by brain size, as additionally removing the effect of intracranial volume (ICV) from cluster AI's had a negligible effect on their interrelations (max correlation change=0.009). Post-hoc tests confirmed that opposite-direction areal asymmetries were more correlated if closer in cortex (Methods); geodesic distance was lower between cluster-pairs that were more correlated (rho=−0.35, p=0.01 [LCBC]; −0.38, p=0.007 [UKB; *Figure 5C*]; −0.34, p=0.02 [HCP]), though this was driven by the aforementioned 'hotspots'. By contrast, same-direction areal asymmetries were not more correlated if closer in cortex (leftward [all p>0.5]; rightward [all p>0.5]). This suggests specific areal asymmetries that are closer in cortex and opposite in direction may show coordinated development.

For thickness asymmetry, the correlation matrix exhibited a clear pattern in UKB that was less visible but still apparent in LCBC and HCP (*Figure 5B*; *Figure 5—figure supplement 1B*). Mantel tests confirmed that the covariance structure replicated between all dataset-pairs (LCBC-UKB r=0.49, p=0.007; LCBC-HCP r=0.62, p<2.9e⁻⁴; UKB-HCP r=0.46, p=0.009). The observed pattern suggested higher leftward asymmetry in regions that are on average left-asymmetric was associated with less rightward asymmetry in regions that are on average right-asymmetric. However, given that the AI measure is bidirectional, closer inspection of the correlations revealed that higher leftward asymmetry in regions that are left-asymmetric actually corresponded to more *leftward* asymmetry in

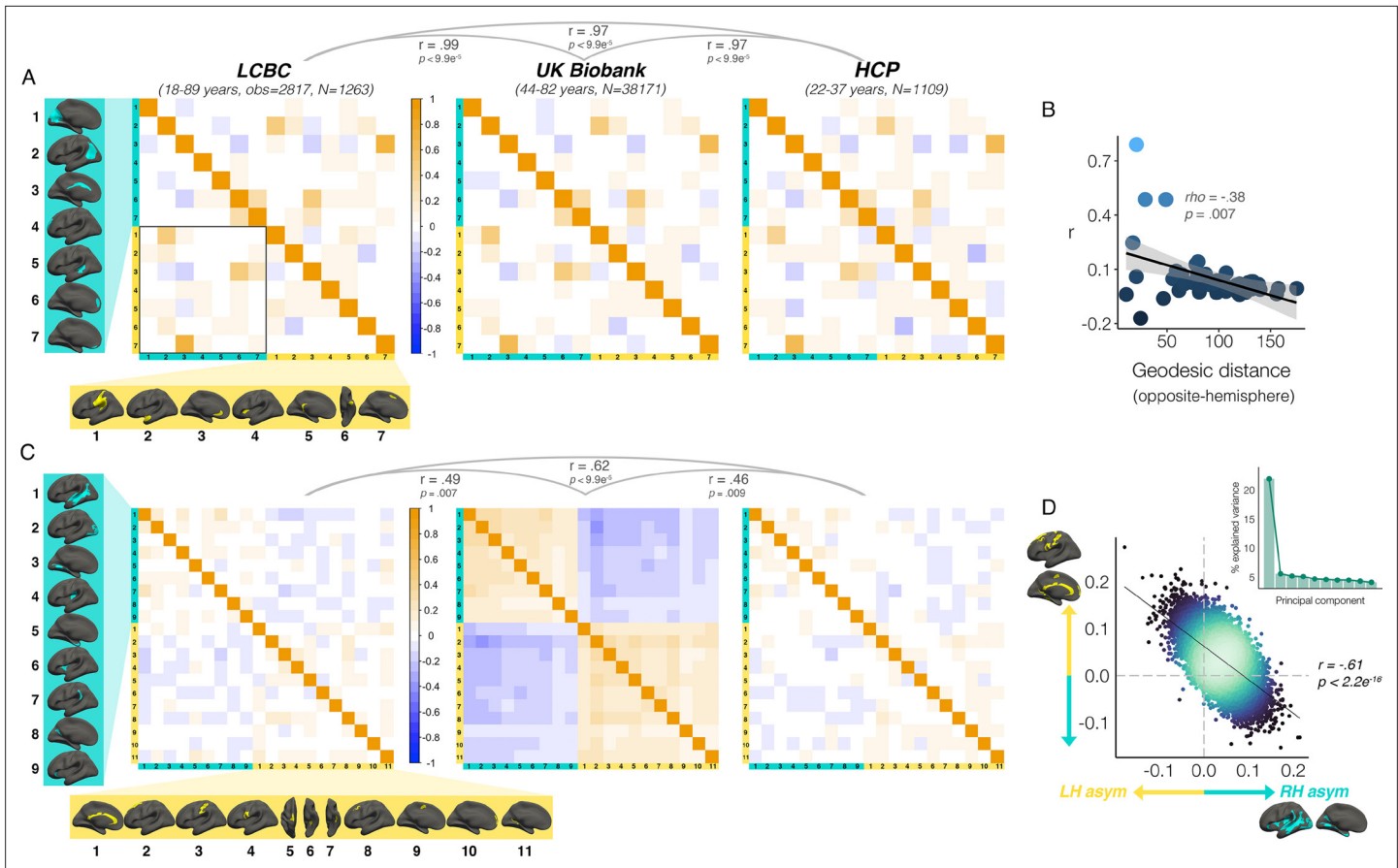

**Figure 5.** Interregional asymmetry correlations. Interregional correlations between (**A**) areal asymmetries and (**B**) thickness asymmetries for each replication dataset (AI's residualized for age, sex, scanner). Individual AI's in rightward clusters are inversed, such that positive correlations reflect positive asymmetry-asymmetry relationships, regardless of direction of mean asymmetry in the cluster (i.e. higher asymmetry in the population-direction). Yellow and blue brain clusters/colours denote leftward and rightward asymmetries, respectively (clusters numbered for reference). A consistent covariance structure was evident both for areal ($r \geq 0.97$) and thickness asymmetry ($r \geq 0.46$; results above matrices). Black box in A highlights relationships between opposite-direction asymmetries (i.e. leftward vs rightward regions). (**C**) For areal asymmetry, asymmetry in opposite-direction cluster-pairs that were closer in cortex was more positively correlated (datapoints show cluster-pairs; geodesic distance in mm). (**D**) A single component explained 21.9% variance across thickness asymmetries in UKB (inset plot). Accordingly, we found a correlation of $r=-0.61$ ($p<2.2e^{-16}$) in UKB between mean asymmetry across leftward clusters (Y-axis) vs. mean asymmetry across rightward clusters (X-axis; AI's in rightward clusters inversed). Lines of symmetry (0) are in dotted grey (see also *Figure 5—figure supplements 1–3*).

The online version of this article includes the following figure supplement(s) for figure 5:

**Figure supplement 1.** Annotated covariance matrices.

**Figure supplement 2.** UK Biobank lower quadrant.

**Figure supplement 3.** Global thickness asymmetry relationships.

**Figure supplement 4.** HCP outliers discarded.

right-asymmetric regions, and vice versa (and on average; see *Figure 5—figure supplement 2*). In other words, individuals may tend towards either leftward lateralization or rightward lateralization (or symmetry) on average, irrespective of the region-specific direction of mean thickness asymmetry in the cluster. Similarly, asymmetry in left-asymmetric regions was mostly positively correlated, and asymmetry in right-asymmetric regions was mostly positively correlated. Again, additionally removing ICV-associated variance had negligible effect (max correlation change=0.01). Post-hoc principal components analysis (PCA) in UKB revealed PC1 explained 21.9% of the variance in thickness asymmetry and suggested a single component may be evident for thickness asymmetry (*Figure 5D*). Accordingly, we found a correlation of $r=-0.61$ between mean asymmetry across all leftward vs. mean asymmetry across all rightward clusters in UKB $p<2.2e^{-16 \text{ [means weighted by cluster size]}}$; see *Figure 5D*; $r=-0.56$,

$p<2.2e^{-16}$ [unweighted means]; $r=0.66$, $p<2.2e^{-16}$ [PC1 across all leftward vs. PC1 across all rightward]. Although less strong, all relationships were significant in LCBC ($r=-0.12$; $p=7.4e^{-11}$ [weighted]; $r=-0.05$; $p=0.005$ [unweighted]; $r=0.19$; $p<2.2e^{-16}$ [PC1 vs. PC1]) and significant or trend-level in HCP ($r=-0.11$; $p=1.6e^{-4}$ [weighted]; $r=-0.04$; $p=0.15$ [unweighted]; $r=0.12$, $p=3.3e^{-5}$ [PC1 vs. PC1]; see **Figure 5—figure supplement 3**). These results suggest thickness asymmetry may be globally interrelated across the cortex and show high directional variability in the adult population.

## Heritability

Although heritability of the global measures for each hemisphere was high (area $h^2_{SNP}$ ~67%, $h^2_{twin}$ ~92%; thickness $h^2_{SNP}$ ~36%, $h^2_{twin}$ ~81%), heritability of global asymmetry measures was substantially lower (area $h^2_{SNP}=7\%$ [95% CI=3–10%], $h^2_{twin}=16\%$ [5–27%]; thickness $h^2_{SNP}=1\%$ [0–5%], $h^2_{twin}=16\%$ [5–27%]). Except the SNP-based estimate for global thickness asymmetry ($P=.22$), all estimates for global measures were post-corrected significant ($p<0.008$; see **Supplementary file 1G**). Of our robust asymmetry clusters, four areal asymmetries showed significant twin-based heritability in HCP (post-correction; leftward SMG/perisylvian and anterior insula, and rightward cingulate and STS), all of which were also significant in the SNP-based analyses ($h^2_{twin}$ range = 19–27%; $h^2_{SNP}$ range = 8–19%). Additionally, SNP-based analyses revealed 9/14 (64%) areal asymmetry clusters exhibited significant heritability (post-correction; **Supplementary file 1H**). Importantly, highest SNP-based heritability was observed for leftward areal asymmetry in the anterior insula cluster ($h^2_{SNP}$ = 18.6%, $p<1.78e^{-15}$; see

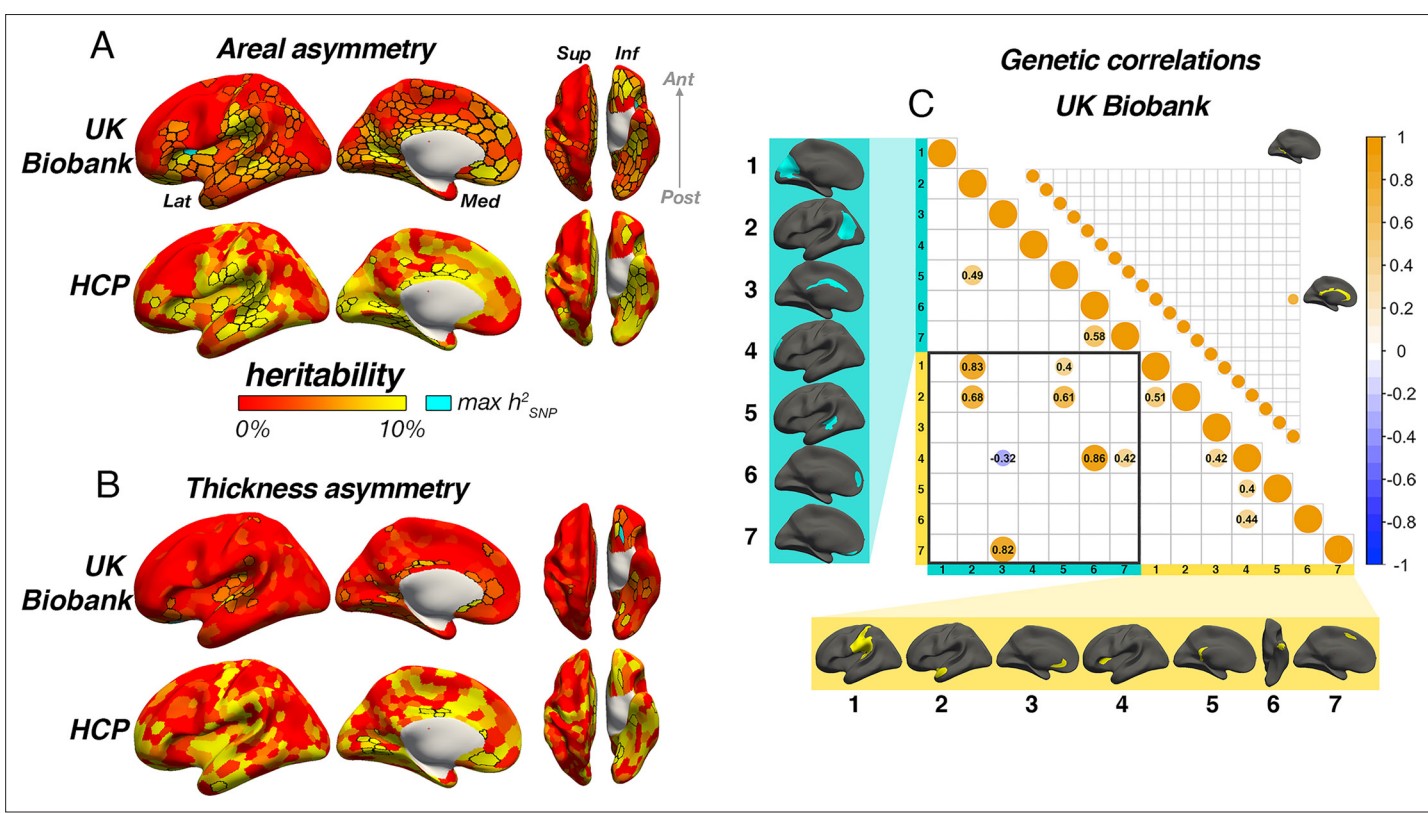

**Figure 6.** Asymmetry heritability. Heritability of areal (**A**) and thickness asymmetry (**B**) estimated cortex-wide using SNP-based (UKB data; top rows) and twin-based methods (HCP data; bottom rows). Unthresholded effect maps are shown. Parcels in black outline show significance at p[FDR]<.05. Cyan parcels depict the point of maximum observed SNP-heritability (area $h^2$=16.4%; thickness $h^2$=16.6%). (**C**) Significant SNP-based genetic correlations (FDR-corrected) between areal (lower matrix) and thickness asymmetries (upper matrix). For area, SNP-based genetic correlations explained several phenotypic correlations (**Figure 5A**). For thickness, one pair survived FDR-correction (shown). See **Figure 6—figure supplement 1** for comparison with genetic correlation estimates from the twin-based HCP sample. Individual AI's in rightward clusters are inversed, such that positive genetic correlations reflect asymmetry-asymmetry genetic relationships, regardless of direction of mean asymmetry in the cluster (i.e. higher asymmetry in the population-direction). Yellow and blue brain clusters/colours denote leftward and rightward asymmetries, respectively (clusters numbered for reference).

The online version of this article includes the following figure supplement(s) for figure 6:

**Figure supplement 1.** Genetic correlations exhibiting pre-corrected significance (p<0.05).

*Supplementary file 1H*), which was substantially higher than the next highest estimates in SMG/perisylvian ($h^2_{SNP}$ = 10.7%, p=3.01e$^{-9}$), retrosplenial cortex, gyrus rectus and the cingulate (all $h^2_{SNP}$ = 8–10%). For thickness, two robust asymmetries survived correction in the twin-based analyses (rightward STS and lingual gyrus). Of these, only STS was also significant in the SNP-based analysis but did not survive correction. Moreover, only 3/20 (15%) thickness asymmetries exhibited significant SNP-based heritability ($h^2_{SNP}$ = 3–7%; *Supplementary file 1I*). Cluster-wise heritability estimates were significantly lower for thickness asymmetry than for area using a SNP-based approach ($\beta$=–1.1, p=0.0008) but not a twin-based approach ($\beta$=–0.03, p=0.33).

We then estimated asymmetry heritability cortex-wide (*Schaefer et al., 2017*; *Figure 6*). For areal asymmetry, 69 parcels survived FDR-correction in HCP, 84% (58) of which were also FDR-corrected significant in the SNP-based analysis. Moreover, a total of 267 (53%) parcels exhibited significant FDR-corrected SNP-based heritability for areal asymmetry in UKB data (significant *p[FDR]*<0.05 parcels in each sample are depicted with black outlines in *Figure 6A*). Beyond significance, a consistent heritability pattern for areal asymmetry was clearly evident using SNP- and twin-based data from independent samples, with higher heritability notably in anterior insula, SMG, Sylvian fissure, STS, calcarine sulcus, cingulate, medial and orbitofrontal cortex, and fusiform. This overlap was substantiated by a spatial correlation of *r*=0.46 between maps; p<2.2e$^{-16}$. Moreover, maximum SNP-heritability (cyan parcel in *Figure 6*) was observed in anterior insula (parcel $h^2_{SNP}$ = 16.4%; p<1.78e$^{-15}$), confirming this region constitutes the most heritable cortical asymmetry in humans (and not improving on the cluster-based estimate). For thickness asymmetry, 15 parcels survived FDR-correction in the twin-based analysis, 7 of which were also post-corrected significant in the SNP-based analysis, and these were typically low estimates (*Figure 6B*). Moreover, significant FDR-corrected SNP-heritability was observed in only 11% (57) of parcels, including around superior temporal gyrus, planum temporale, the posterior insula/Sylvian fissure, anterior insula, and in orbitofrontal cortex (max $h^2_{SNP}$ = 16.6%), along the cingulate and in medial visual cortex. Beyond significance, we observed little obvious visual overlap in twin- and SNP-based heritability patterns (spatial correlation was significant but low; *r*=0.24; p=0.01), and higher estimates pertained to regions that were limited in extent but generally showed no clear global pattern common to both datasets, with the possible exception of calcarine, cingulate and orbitofrontal cortex. Furthermore, cortex-wide heritability estimates were significantly lower for thickness asymmetry than for area using both a SNP-based ($\beta$=–0.71, p<2.2e$^{-16}$) and twin-based approach ($\beta$=–0.33, p=1.08e$^{-7}$).

For areal asymmetry, large SNP-based genetic correlations explained several phenotypic correlations evident in *Figure 5A* (*Figure 6C–D*; Methods). For example, high SNP-based genetic correlations were found between leftward asymmetry in SMG/perisylvian and higher rightward asymmetry in lateral parietal cortex (LPC; $rG_{SNP}$ = 0.83; *p[FDR]*=6.58e$^{-5}$), between leftward superior frontal cortex asymmetry and rightward asymmetry along the cingulate ($rG_{SNP}$ = 0.82; *p[FDR]*=0.01), and between leftward anterior temporal/parahippocampal asymmetry and rightward asymmetry in LPC ($rG_{SNP}$ = 0.68; *p[FDR]*=0.01). SNP-based genetic correlations between anterior insula and two rightward superior frontal clusters were also observed ($rG_{SNP}$ = 0.86; *p[FDR]*=1.21e$^{-6}$; $rG_{SNP}$ = 0.42; *p[FDR]*=6.64e$^{-4}$) in the absence of phenotypic correlations (*Figure 5*), and several same-direction asymmetries showed moderate genetic correlation. In contrast, only one post-corrected significant SNP-based genetic correlation was found for thickness asymmetry (see *Figure 6C*; $rG_{SNP}$ = 0.68; *p[FDR]*=0.03). We tested replication in the twin-based HCP sample across all cluster-pairs tested in the SNP-based analysis (Methods; note not all of these exhibited significant twin-based heritability; *Supplementary file 1H-I*). Two twin-based genetic correlations survived FDR-correction, both for areal asymmetry (*Figure 6—figure supplement 1*). One was the high genetic correlation between leftward SMG/perisylvian and rightward LPC areal asymmetry observed in the SNP-based analysis ($rG_{twin}$ = 1.0 [95% CI=0.69–1.0], *p[FDR]* = 0.03). The second was between leftward anterior insula and leftward SMG/perisylvian areal asymmetry and was not post-corrected significant in the SNP-based analysis ($rG_{twin}$ = 0.59 [0.24–1.0], *p[FDR]*=0.03). Of note, while several areal asymmetry cluster-pairs exhibiting post-corrected significance in the SNP-based analysis were estimated to show high twin-based genetic correlation, these did not survive correction (see *Figure 6—figure supplement 1*). For thickness asymmetry, no twin-based genetic correlation survived correction, and there was little overlap in estimates produced betweeen methods.

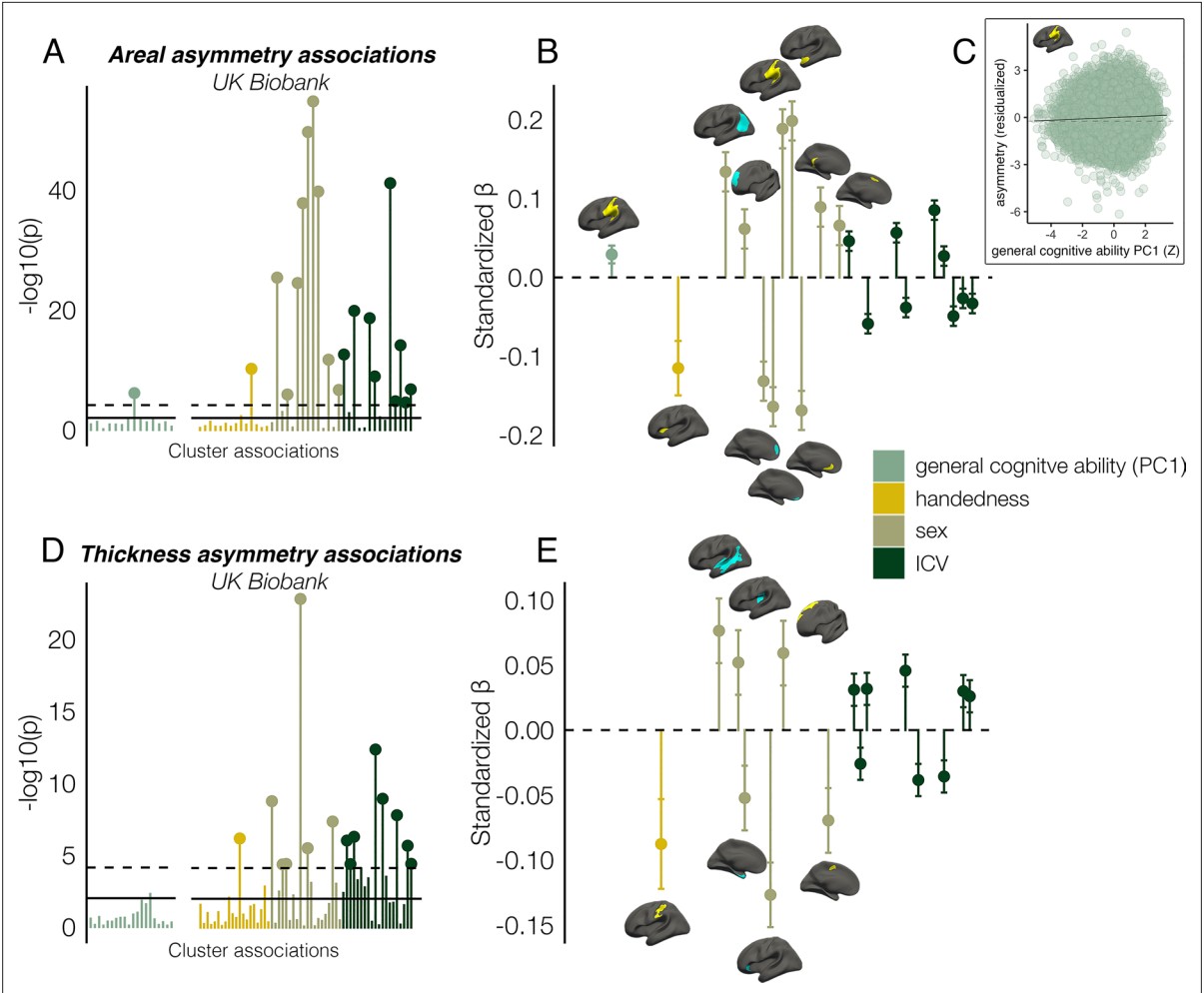

**Figure 7.** Asymmetry associations with general cognitive ability (first principal component [PC1]), handedness, sex, and intracranial volume (ICV) in UKB, in clusters exhibiting population-level areal (upper) and thickness asymmetry (lower). (**A, D**) Significance of associations (negative logarithm; corrected [p<7.4e$^{-5}$] and uncorrected threshold [p=0.01] shown by dotted and non-dotted line, respectively). X-axis displays the test for each cluster-association. As maximum sample size was used to test each association, effects of general cognitive ability were tested in separate models with fewer observations (N=35,198; separated association plots) than handedness, sex and ICV (N=37,569). (**C**) Visualization of the found association between leftward areal asymmetry in the large supramarginal cluster with general cognitive ability. The line of null association is shown for comparison (dotted) (**B, E**) Right plots denote effect sizes, 95% confidence intervals (error bars) and cortical location of associations surpassing Bonferroni-corrected significance. Individual AIs in rightward clusters were inversed. Right handers and females are coded 0, such that a negative effect for general cognitive ability / handedness / sex / ICV / reflects less asymmetry in higher cognition / left handers / males / larger brains. Associations with ICV are shown in **Figure 7— figure supplement 2**. Yellow and blue clusters denote leftward and rightward asymmetries, respectively.

The online version of this article includes the following figure supplement(s) for figure 7:

**Figure supplement 1.** Cognitive associations.

**Figure supplement 2.** ICV effects (continuation of **Figure 7**).

## Associations with cognition, handedness, sex, and ICV

Finally, we found several significant associations between asymmetry and factors purportedly related to it in UKB data (**Figure 7**; **Supplementary file 1J-K**). Notably, all effect sizes were small. For general cognitive ability, we found one association, wherein higher areal asymmetry in the largest leftward cluster (SMG/perisylvian) was significantly associated with better cognition ($\beta$=0.03 [CI = 0.02 –0.04], p=7.4e$^{-7}$, **Figure 7C**). Although small, we note this association was far from only just surviving correction at our predefined alpha level ($\alpha$=0.01; 136 tests; Methods). This was checked in the substantially reduced non-imputed subset of data with no missing cognitive variables and retained the lowest p-value (N=4696; $\beta$=0.04 [CI = 0.01 -0.07]; p=6.9e$^{-3}$). We also post-hoc tested areal asymmetry

associations with the 11 separate cognitive tests (*Figure 7—figure supplement 1*). For handedness, reduced leftward areal asymmetry in anterior insula and thickness asymmetry along postcentral gyrus was evident in left-handers, in line with our recent vertex-wise mapping in UKB (*Sha et al., 2021a*). For sex effects, which were also small, males typically exhibited slightly stronger areal asymmetry in large clusters (e.g. leftward SMG/perisylvian and temporal pole; rightward inferior parietal and superior frontal) but reduced leftward and rightward asymmetry in mPFC. For thickness, males exhibited more rightward asymmetry in STS and posterior insula, more leftward thickness asymmetry in superior frontal cortex, but reduced rightward thickness asymmetry in entorhinal cortex and anterior insula, and reduced leftward asymmetry in caudal superior frontal cortex. As ICV effects were typically most nominal, these are shown in *Figure 7—figure supplement 2*. None of the reported associations changed appreciably when controlling for additional brain-size related covariates (*Williams et al., 2022*; *Supplementary file 1J-K*).

## Discussion

Combining the strengths of a vertex-wise delineation of population-level cortical asymmetry in 7 international datasets and dense longitudinal data, we offer the first description of the longitudinal developmental and lifespan trajectories of cortical asymmetry, advancing knowledge on normal human brain development. We show areal asymmetry is predominantly stable across life, whereas we trace developmental growth in many thickness asymmetries signifying differentiation of thickness asymmetry from early childhood to the mid-20s. We further demonstrate the replicable interregional relationships between asymmetries within individuals, provide the most detailed heritability maps for cortical asymmetry to date, and uncover novel and confirm previously-reported associations with factors purportedly related to asymmetry – all with small effects. All maps are available in *Supplementary file 2*.

### Population-level asymmetry

Our vertex-wise analysis of cortical asymmetries that reproduce across adult cohorts replicates and completes a recent low-resolution meta-analysis (*Kong et al., 2018*), and can serve as a high-fidelity phenotype for future brain asymmetry studies. The marked consistency across samples suggests consensus may now be reached regarding cortical asymmetry phenotypes in humans, as our results agree with most reported results for areal asymmetry (*Kong et al., 2018*; *Chiarello et al., 2016*; *Lyttelton et al., 2009*; *Maingault et al., 2016*; *Li et al., 2014*), as well as several, typically more recent reports for thickness asymmetry (*Kong et al., 2018*; *Roe et al., 2021*; *Li et al., 2015*; *Plessen et al., 2014*; but see *Figure 1—figure supplement 4* and Limitations for an outstanding issue regarding the biological origin of thickness asymmetry). Indeed, for thickness asymmetry – for which findings have been particularly mixed (*Kong et al., 2018*; *Chiarello et al., 2016*; *Roe et al., 2021*; *Li et al., 2015*; *Luders et al., 2006*; *Zhou et al., 2013*; *Maingault et al., 2016*; *Shaw et al., 2009*; *Lou et al., 2020*; *Hamilton et al., 2007*; *Zhou et al., 2018*; *Koelkebeck et al., 2014*; *Plessen et al., 2014*) – the left-right patterning here is compatible with low-resolution meta-analyses (*Kong et al., 2018*), asymmetries evident in the first months of life (*Li et al., 2015*; *Williams et al., 2023*), a large-scale mapping in mid-old age (*Sha et al., 2021a*), reports using alternative analysis streams *Li et al., 2015*; *Plessen et al., 2014*, and possibly the overall pattern of brain torque (*Kong et al., 2018*; *Hugdahl, 2011*) – a gross hemispheric twist leading to frontal and occipital bending at the poles (*Toga and Thompson, 2003*). The high overlap in effects between seven datasets from four countries here suggests these results likely apply universally. This evident consensus suggests genetic-developmental programs regulate mean brain lateralization with respect to both area and apparent thickness in humans. However, the genetic findings presented herein suggest these may have reached population fixation, as heritability of even our optimally delineated asymmetry measures was generally low. This indicates either subject-specific stochastic mechanisms in early neurodevelopment or later developmental influences primarily determine cortical asymmetry. Tracing their lifespan development, we show the trajectories of areal asymmetry primarily suggest this form of asymmetry is developmentally stable at least from age ~4, maintained throughout life, and formed early on – possibly in utero (*Sha et al., 2021b*; *Li et al., 2014*; *Williams et al., 2023*) (while we cannot extrapolate to ages before our sample begins, we note this agrees with findings in neonates *Li et al., 2014*; *Williams et al., 2023*). One

interpretation of lifespan stability combined with low heritability may be stochastic early-life developmental influences determine individual differences in areal asymmetry more than later developmental change, but work linking prenatal and childhood trajectories is needed to affirm this. Still, we also found relatively stronger heritability for areal asymmetry (notably, anterior insula exhibited ~19% SNP-heritability). This also illustrates region-dependent genetic effects upon areal asymmetry, and high genetic correlations suggest specific areal asymmetries are formed under common genetic influence. In contrast, childhood development of thickness asymmetry until a peak around age ~24 (*Roe et al., 2021*), higher directional variability in adult samples, and lower heritability all converge to suggest thickness asymmetry may be more shaped through subject-specific effects in later childhood, possibly through interaction with the environment. This interpretation applied to asymmetry also agrees with work suggesting cortical area in general may trace more to early-life factors (*Walhovd et al., 2016*; *Grasby et al., 2020*; *Rakic, 1995*) whereas thickness may be more impacted by lifespan influences (*Grasby et al., 2020*; *Fjell et al., 2019*).

## Lifespan trajectories

Our longitudinal description of cortical asymmetry lifespan trajectories gleaned novel insight into normal brain maturation. For areal asymmetry, adult-like patterns of lateralization were strongly established before age ~4, indicating areal asymmetry traces back further and does not primarily emerge through later cortical expansion (*Wierenga et al., 2014*). Rather, the lifespan trajectories predominantly show stability from childhood to old age, as asymmetry was maintained through periods of developmental expansion and aging-related change that were region-specific and bilateral. This may align with evidence indicating areal asymmetry may be primarily determined in utero (*Li et al., 2014*; *Williams et al., 2023*), including evidence suggesting little change in areal asymmetry from birth to 2 years (*Li et al., 2014*; *Li et al., 2013*; *Wierenga et al., 2014*), and little difference between maps derived from neonates and adults (*Li et al., 2014*; *Williams et al., 2023*). It may also fit with the principle that the primary microstructural basis of cortical area (*Rakic, 1995*) – the number of and spacing between cortical minicolumns – is determined in prenatal life (*Rakic, 1995*; *Fjell et al., 2019*), and agree with work suggesting asymmetry at this microstructural level may underly hemispheric differences in surface area (*Chance et al., 2006*). The developmental trajectories agree with studies indicating areal asymmetry is established and strongly directional early in life (*Li et al., 2014*; *Remer et al., 2017*). That change in surface area later in development follows embryonic gene expression gradients may also agree with a prenatal account for areal asymmetry (*Fjell et al., 2019*). Our results may therefore constrain the extent to which areal asymmetry can be viewed as a plastic feature of brain organization, and may even suggest areal asymmetry may sometimes be a marker for innate hemispheric specializations shared by most humans. Although future research is needed to assess structure-function relationships, the degree of precision with which leftward areal asymmetry follows the contours of auditory regions in the Sylvian fissure (*Figure 1—figure supplement 1*) that show left functional lateralization may be one example (*Chance et al., 2006*; *Tzourio-Mazoyer et al., 2018*; *Ocklenburg et al., 2018*); we found ~94% of individuals exhibited leftward areal asymmetry here – the most consistently lateralized cortical region in humans (*Figure 1F*).

In contrast, although weak thickness asymmetry was evident by age ~4, we observed childhood developmental growth in many thickness asymmetries. Developmental trajectories showed non-linear asymmetry growth by virtue of accelerated thinning of the non-dominant hemisphere (10/20 clusters showed this relative hemispheric difference in the rate of developmental thinning; *Figure 4*). This led to maximally established asymmetry around ~24 years of age. These trajectories clearly suggest differentiation of the cortex occurs with respect to thickness asymmetry in development, possibly (though not necessarily) suggesting it may be more amenable to experience-dependent change. Indeed, as cortical thinning in childhood is thought to partly reflect learning-dependent processes such as intracortical myelination (*Natu et al., 2019*), pruning of initially overproduced synapses (*Petanjek et al., 2011*; *Faust et al., 2021*) and neuropil reduction, thickness asymmetry growth may suggest hemispheric differences in the developmental optimization of cortical networks at least partly shaped by childhood experience. This raises the possibility thickness asymmetry may be a marker of ontogenetic hemispheric specialization within neurocognitive networks, possibly in line with animal models suggesting lateralized training can alter the balance in homotopic thickness (*Díaz et al., 1994*). However, this is speculative, and intervention approaches are needed to experimentally test

plasticity of thickness asymmetry in humans. Our lifespan results are difficult to reconcile with earlier lifespan reports finding different mean thickness asymmetry patterns (*Zhou et al., 2013*; *Shaw et al., 2009*; *Zhou et al., 2018*) and modelling asymmetry linearly across cross-sectional childhood and adult samples (*Zhou et al., 2013*; *Zhou et al., 2018*; *Plessen et al., 2014*). However, our findings agree with work finding a similar left-right thickness asymmetry pattern shows rapid longitudinal increase in the first years of life (*Li et al., 2015*), particularly in mPFC (*Li et al., 2015*). As we observed rapid asymmetry differentiation spanninng childhood and adolescence in most prefrontal regions and others (*Figure 3*; *Figure 4*; *Figure 3—figure supplements 3 and 4*), we extend these earlier findings in neonates (*Li et al., 2015*). Likely, longitudinal data and nonlinear modelling was critical to capture early developmental growth in thickness asymmetry within a lifespan perspective, because large variation in hemispheric thickness estimates at any age may hinder detection of subtle change effects even in large cross-sectional samples (*Fjell et al., 2020*), rendering follow-up data likely critical in the context of cortical asymmetry change. As prefrontal thickness asymmetry seems particularly vulnerable in some neurodevelopmental disorders (*Postema et al., 2019*), aging, and Alzheimer's disease (*Roe et al., 2021*), these trajectories provide a useful normative reference regarding the timing of expected brain change in development. With regards to aging, most clusters delineated here did not exhibit the relative hemispheric difference in rates of cortical thinning we have previously shown is a feature of aging in heteromodal cortex (*Roe et al., 2021*), except in clusters overlapping with our previous analysis (*Figure 3—figure supplement 5*). This fits with our previous work showing aging-related loss in thickness asymmetry is specific to heteromodal regions vulnerable in aging. In these, we also found strong evidence of early developmental growth in thickness asymmetry (*Figure 3—figure supplement 4*). Developmental differentiation and aging-related dedifferentiation of thickness asymmetry underscores its proposed role in supporting optimal brain organization and function, though it seems not all thickness asymmetries that grow in childhood development decline in aging.

## Interregional asymmetry correlations

For areal asymmetry, we uncovered a covariance structure that almost perfectly replicated across datasets. In general, this fit with a multifaceted view (*Bain et al., 2019*; *Rentería, 2012*; *Francks, 2015*), in which most asymmetries were either not or only weakly correlated – but reliably so – contrasting views emphasizing a single biological (*Annett, 1998*; *Annett, 1964*) or overall anatomical factor (*Crow, 2010*) controlling cerebral lateralization. However, we also identified several regions wherein areal asymmetry reliably correlated within individuals, showing the variance in cortical asymmetries is not always dissociable, as often thought (*Bain et al., 2019*; *Rentería, 2012*; *Francks, 2015*). The strongest relationships all pertained to asymmetries that were proximal in cortex but opposite in direction. Several of these were underpinned by high asymmetry-asymmetry SNP-based genetic correlations, illustrating some lateralizations in surface area exhibit coordinated genetic development.

For thickness asymmetry, we also uncovered a common covariance structure – particularly clear in UKB – that nevertheless replicated with moderate precision across datasets (*Figure 5C*). Furthermore, a single component explained 21.9% variance in thickness asymmetry in UKB, and a high correlation across 38,171 individuals further suggested thickness asymmetry may be globally interrelated across the cortex (*Figure 5D*). These data for thickness indicate individuals may tend towards either leftward asymmetry, rightward asymmetry, or symmetry, both globally across the cortex and irrespective of the region-specific average direction of asymmetry (*Figure 5—figure supplements 2–3*). Though it is unclear why the relationships were weaker in the other datasets, we nevertheless found similarly significant relationships in each (*Figure 5—figure supplement 3*). This result may be in broad agreement with the notion that some lateralized genetic-developmental programs may trigger lateralization in either direction or lose their directional bias through environmental interaction (*Francks, 2015*). As thickness asymmetry seems established at but minimal from birth (*Li et al., 2015*), genetic effects may determine the average region-specific hemispheric bias in the population, but later developmental change may subsequently confer major increases upon its directional variance (*Francks, 2015*). Overall, our data suggests developmental change in thickness asymmetry may lead to directional variability of its lateralization across individuals. Thus, far from being independent phenotypes (*Bain et al., 2019*; *Rentería, 2012*), thickness asymmetries may be globally interrelated across cortex and their direction coordinated through childhood.

## Genetic influences

For areal asymmetry, we found replicable patterns of low-moderate heritability across datasets and methods. We also found areal asymmetry in anterior insula is, to our knowledge, the most heritable asymmetry yet reported with genomic methods (*Sha et al., 2021a*; *Sha et al., 2021b*; *Carrion-Castillo et al., 2020*; *Cuellar-Partida et al., 2021*; *Elliott et al., 2018*), with common SNPs explaining ~19% variance. This is notably higher than in our recent report (<5%) (*Sha et al., 2021a*), illustrating a benefit of our approach. As we reported recently (*Sha et al., 2021a*), we confirm asymmetry here associates with handedness. Furthermore, highest SNP and twin-based heritability for areal asymmetry was found in regions constituting the earliest emerging cortical asymmetries in utero (*Dubois et al., 2010*; *Habas et al., 2012*; *Kasprian et al., 2011*; *Hill et al., 2010*): anterior insula, STS, PT, medial occipital cortex, and parahippocampal gyrus (*Figure 6A*). However, heritability was not restricted to these regions, as most areal asymmetries exhibited significant – albeit often lower – SNP-based heritability, as did most parcels when estimated cortex-wide. SNP-based heritability was also evident in regions not found in the present analyses to show strong areal asymmetry, such as Broca's area (but see *Figure 1—figure supplement 5*). The effects agree with and elaborate on two previous genetic explorations (*Kong et al., 2018*; *Sha et al., 2021b*) and reports of heritable areal asymmetry in handedness-associated clusters (*Sha et al., 2021a*). However, while heritability of either hemisphere was high, areal asymmetry heritability was still only moderate at best, suggesting genetic but primarily subject-specific stochastic effects underly its formation. By contrast, thickness asymmetry was generally not heritable, or showed low and localized heritability effects with no clear global pattern, as well as more divergent results using twin and genomic methods, possibly due to low-power for twin-models.

Together, lifespan stability (*Li et al., 2014*), higher heritability, and phenotypic and genetic correlations suggest higher genetic influence upon individual differences in areal asymmetry. By contrast, childhood developmental growth, directional variability and low heritability suggest thickness asymmetry may be more shaped through individual lifespan exposures (*Fjell et al., 2019*). Whether region-specific thickness asymmetry change relates to the maturation of lateralized brain functions is thus an interesting question for future research (*Bishop and Bates, 2019*; *Somers et al., 2015*). Regardless, these results support a differentiation between early-life (i.e. before age ~4) and later developmental factors in shaping areal and thickness asymmetry, respectively.

## Individual differences

Other factors commonly espoused to be important for asymmetry were associated with only small average effects in adults. For example, we found one region – SMG/perisylvian – wherein higher leftward areal asymmetry related to subtly higher cognitive ability. Since interhemispheric anatomy here is likely related to brain torque (*Toga and Thompson, 2003*; *LeMay, 1976*), this may agree with work suggesting torque relates to cognitive outcomes (*Kong, 2019*; *Zhao et al., 2021*). Interestingly, that ~94% of humans exhibit leftward asymmetry in this region (*Figure 1G*) suggests tightly regulated genetic-developmental programs control its lateralized direction in humans (see *Figure 6*). This result may therefore suggest disruptions in areal lateralization early in life are associated with cognitive deficits detectable in later life as small effects in big data (*Deary et al., 2004*). While speculative, this may also agree with evidence that differences in general cognitive ability that show high lifespan stability (*Deary et al., 2004*) relate primarily to areal phenotypes formed early in life (*Walhovd et al., 2016*; *Rakic, 1995*; *Fjell et al., 2019*).

Consistent with our recent analysis in UKB (*Sha et al., 2021a*), we confirmed leftward areal asymmetry of anterior insula, and leftward somatosensory thickness asymmetry is subtly reduced in left-handers. Sha et al. (*Sha et al., 2021a*) reported shared genetic influences upon handedness and asymmetry in anterior insula and other more focal regions. Anterior insula lies within a left-lateralized functional language network (*Labache et al., 2019*), and its structural asymmetry may relate to language lateralization (*Chiarello et al., 2013*; *Biduła and Króliczak, 2015*; *Keller et al., 2011*) in which left-handers show increased atypicality (*Wiberg et al., 2019*; *Mazoyer et al., 2014*; *Westerhausen et al., 2006*; *Carey and Johnstone, 2014*). Since asymmetry here emerges early in utero (*Dubois et al., 2010*) and is by far the most heritable (*Figure 6*), we agree with others (*Chiarello et al., 2013*) that this ontogenetically foundational region of cortex may be fruitful for understanding genetic-developmental mechanisms influencing laterality (*Afif et al., 2007*; *Kalani et al., 2009*). Less leftward somatosensory thickness asymmetry in left-handers also echoes our recent report (*Sha et al., 2021a*)

and fits a scenario whereby thickness asymmetries may be partly shaped through use-dependent plasticity and detectable through group-level hemispheric specializations of function. Still, the small effects show cortical asymmetry cannot predict individual handedness. Associations with other factors typically assumed important were similarly small, and mostly compatible with the ENIGMA report (*Kong et al., 2018*) and elsewhere (*Williams et al., 2022*; *Guadalupe et al., 2015*). Concerning sex effects – which were small, differing in direction, and more predictive than ICV (*Guadalupe et al., 2015*) – inconsistencies between ours and ENIGMA include findings of increased (here) and decreased (*Kong et al., 2018*) LPC areal asymmetry in males, and increased (*Kong et al., 2018*) and decreased (here) entorhinal thickness asymmetry in males, and our approach detected other regions slightly more asymmetric in males (e.g. STS). Possibly, differences in sample median age (UKB = ~64; *Kong et al., 2018* = 26) and potential sex-differences in brain decline (*McCarrey et al., 2016*) may underlie some inconsistencies.

## Limitations

Some limitations should be noted. First, our delineation of population-level asymmetry used a single analysis software. We used FreeSurfer's default 'recon-all' to delineate the cortex, which has been extensively validated against postmortem data (*Cardinale et al., 2014*) and is the software underlying most large-scale studies with brain measures. It is currently unclear to what extent differences in pipelines account for previous mixed results. Although we highlight there are clear commonalities between our results and studies using alternative pipelines, suggesting they generalize across analysis streams (*Li et al., 2015*; *Luders et al., 2006*; *Lyttelton et al., 2009*; *Plessen et al., 2014*), we found one instance where the MRI pipeline leads to different results for thickness asymmetry (HCP pipeline; *Figure 1—figure supplement 4*). It is not known what underlies this difference, though it is unrelated to the cross-hemispheric registration methods employed here, as our results reproduce using standard methods and thus are likely evident in most FreeSurfer-derived datasets (*Figure 1—figure supplement 5*). One possibility could be that thickness asymmetry may not reflect cortical thickness differences per se, but rather reflect biologically meaningful hemispheric differences in intracortical myelination that are consistently picked up on via FreeSurfer's delineation procedure. Future research is required to resolve this. However, that the thickness asymmetry pattern we observe shows a clear developmental trajectory suggests it is a true biological effect (*Figure 3*; *Figure 4*; *Figure 3—figure supplements 1–4*), as do the replicable aging-related changes therein we have shown previously (*Roe et al., 2021*). Other sources of inconsistently reported results for cortical asymmetries likely include varying age-distributions (*Roe et al., 2021*), and multiple asymmetries within atlas-based parcels (e.g. we observed discrepant results to ENIGMA for insula thickness asymmetry *Kong et al., 2018*). Still, this does not explain other discrepancies, such as rightward areal STS asymmetry here but not elsewhere (*Kong et al., 2018*; *Bain et al., 2019*; *Remer et al., 2017*; *Figure 1—figure supplement 2*). Thickness asymmetries seem also more variable compared with area (*Kong et al., 2018*), though it should be noted thickness in general may be slightly less reliable (*Hedges et al., 2022*), the asymmetry effect is smaller, and thus possibly contains more error. Relatedly, while areal and thickness asymmetry patterns and magnitudes using cross-hemispheric methods agree with standard analysis (*Figure 1—figure supplement 5*), the magnitude of some asymmetries near the subcortical boundary may be exaggerated via this approach (*Sha et al., 2021a*). And while we did not find (strong) areal asymmetry in inferior frontal regions (*Kong et al., 2018*), both the unthresholded significance maps and standard parcellation analyses were compatible with this (*Figure 1—figure supplement 2* and *Figure 1—figure supplement 5*), highlighting a limitation of our thresholding approach. Second, while GAMMs are considered optimal for modelling lifespan data and are robust to nonuniform age distributions (*Sørensen et al., 2021*), relative underrepresentation in mid-adulthood may drive trajectory inflection points around this age (*Roe et al., 2021*), urging caution around interpreting these as reflecting real change. Relatedly, as individual-level mean estimates likely contain more error when extracted from smaller clusters, the trajectories from smaller clusters may be more prone to deflection by age density differences. And while spatially averaging across asymmetries helps smooth out some of this noise to better reveal developmental principles underlying cortical asymmetries (*Figures 2–3*), it also trades off with accuracy, since trajectories from distinct regions somewhat differ. Third, although differing heritability methods enabled a replication test, methodological differences should be considered when interpreting heritability estimates, and may partly explain the common observation of

higher estimates from twins, which we also observed. For example, twin methods implicitly incorporate additive genetic effects and gene-gene interactions amongst other terms, whereas SNP-based methods incorporate only additive effects. Still, twin methods may be prone to overestimating heritability due to unmet assumptions (*Dalmaijer, 2020*), whereas SNP-based methods may not capture all phenotype-relevant genetic variance and have their own assumptions (*Hibar et al., 2017*). Adding to this complexity, the samples used for twin- and SNP-based estimation consist of young adults and older adults, respectively. Hence, low SNP-based estimates for thickness asymmetry in UKB may be partly due to reduced thickness asymmetry in older adults (*Roe et al., 2021*; *Williams et al., 2022*). However, as cortex-wide twin-based estimates were also significantly lower for thickness asymmetry in the young HCP sample, age is likely not the main driver behind this difference. Again, we also cannot rule out that reliability differences between measures may partially explain some magnitude differences in heritability. Further, genetic correlations can be high even where heritability of either trait is low, and are typically higher than phenotypic correlations – known phenomena also observed here (*Cheverud, 1988*; *Sodini et al., 2018*). It may therefore be prudent to not overinterpret their magnitudes, though the veracity of the reported SNP-based genetic correlations seems well-supported, as one expects true genetic correlations between developmentally-related traits (here, sampled from nearby in the same organ), and to track the phenotypic correlations (see also *Figure 6—figure supplement 1*). Fourth, we imposed a cluster size limit for overlapping asymmetry effects, and thus more focal asymmetries may also be informative for the factors tested here (*Sha et al., 2021a*). Fifth, only dichotomous handedness self-reports are available with UKB, and future studies might benefit from more nuanced handedness assessments. Relatedly, because UKB cognitive data is not exhaustive (*Fawns-Ritchie and Deary, 2020*), we extracted the common variance across tests to index general cognitive ability. This approach does not permit testing associations with well-operationalized or specific cognitive domains, and it remains to be seen whether cortical asymmetry may be more informative for lateralized cognition (*Ocklenburg et al., 2014*).

Overall, we track the development of population-level cerebral cortical asymmetries longitudinally across life and perform analyses to trace developmental principles underlying their formation. Developmental and lifespan trajectories, interregional correlations and heritability analyses converge upon a differentiation between early-life and later-developmental factors underlying the formation of areal and thickness asymmetries, respectively. By revealing hitherto unknown principles of developmental stability and change underlying diverse aspects of cortical asymmetry, we here advance knowledge of normal human brain development.

## Methods
### Samples
We used anatomical T1 -weighted (T1w) scans from 7 independent international MRI datasets originating from 4 countries (see *Supplementary file 1A* for an overview of samples used for each analysis). Note that with the exception of vertex-wise analyses in UKB (see below), all analyses made use of all available observations from each sample meeting the stated age-range criteria for each analysis. Studies conducted at the Center for Lifespan Changes in Brain and Cognition (LCBC *Vidal-Pineiro et al., 2020*) were approved by the Regional Ethical Committee of South-East Norway (2017/653) and complied with all relevant ethical regulations. Ethical approval for the other datasets was granted by the relevant authorities (*Supplementary file 1M*).

### Reproducibility across samples: population-level asymmetry
To delineate average adult patterns of whole-cortical areal and thickness asymmetry, we restricted the age-range of all samples used in the vertex-wise analyses to 18–55. **Dataset 1:** Here, the LCBC sample comprised 1572 mixed cross-sectional and longitudinal scans (N longitudinal = 812; timepoint range = 1–6) from 923 participants (mean age = 30.6 ± 9.6) collected across 2 scanners. Additionally, 125 individuals were double-scanned at the same timepoint on both scanners. **Dataset 2:** The Cambridge Centre for Ageing and Neuroscience (*Cam-CAN*) (*Shafto et al., 2014*) sample comprised cross-sectional scans of 321 individuals (mean age = 38.7 ± 9.7) (*Taylor et al., 2017*). **Dataset 3:** The Dallas Lifespan Brain Study (*DLBS*) (*Kennedy et al., 2012*) sample comprised cross-sectional scans of 160 individuals (mean age = 37.5 ± 10.7). **Dataset 4:** The Southwest University Adult Lifespan Dataset

(*SALD*) (*Wei et al., 2018*) sample comprised cross-sectional scans of 301 individuals (mean age = 33.7 ± 11.5). **Dataset 5:** The *IXI* sample comprised cross-sectional scans of 313 healthy individuals collected across 3 scanners (mean age = 36.8 ± 9.6; http://brain-development.org/ixi-dataset). **Dataset 6:** Here, the Human Connectome Project (*HCP*) 1200 (*Van Essen et al., 2013*) sample comprised 1111 scans (mean age = 28.8 ± 3.7). **Dataset 7:** Here, the UKB sample consisted of 1000 randomly sampled cross-sectional scans (mean age = 52.1 ± 1.9), restricted to be comparable in size to the other datasets in this analysis.

## Lifespan trajectories

Here, we used the full age-range of the longitudinal lifespan LCBC sample (4.1–89.4 years), 3937 cross-sectional and longitudinal scans (N longitudinal = 2762) from 1886 individuals (females = 1139; mean age = 36.8) collected across 4 scanners (271 double-scans) (*Fjell et al., 2020*; *Vidal-Pineiro et al., 2019*).

## Interregional asymmetry correlations

Here, we used the three largest datasets: LCBC (N=1263; N obs = 2817), UKB (N=38,171), and HCP (N=1109; two outliers removed; see below), excluding the childhood age-range from the LCBC sample not covered in any other dataset.

## Heritability and individual differences

For twin heritability, we used HCP 1200 extended twin data (1037 scans from twins and non-twin siblings; age-range=22–37; mean age = 28.9 ± 3.7). The various kinships are described in *Supplementary file 1B*. All included twin pairs were same-sex. For SNP-heritability, we used the UKB imaging sample with genome-wide data surpassing quality control (N=31,433). For individual differences analyses, we used the UKB imaging sample with the maximum number of available observations for each variable-of-interest (see below).

# MRI preprocessing

T1w anatomical images (see *Supplementary file 1C* for MRI acquisition parameters) were processed with FreeSurfer (v6.0.0) (*Fischl and Dale, 2000*) and vertex-wise areal and thickness morphometry estimates were obtained for each MRI observation. As the LCBC sample also contained longitudinal observations, initial cross-sectional reconstructions in LCBC were subsequently ran through FreeSurfer's longitudinal pipeline. As HCP data was acquired at a higher voxel resolution (0.7 mm isotropic), the T1w scans were processed with the `--hires` flag to recon-all (*Zaretskaya et al., 2018*). Areal and thickness maps of the LH and RH of each participant in each dataset were resampled from the native cortical geometry to a symmetrical surface template ('*LH_sym*') (*Maingault et al., 2016*; *Marie et al., 2016*) based on cross-hemispheric registration (*Greve et al., 2013*). This procedure achieves vertex-wise alignment of the data from each participant and homotopic hemisphere in a common analysis space, enabling a whole-cortical and data-driven analysis of cortical asymmetry. Areal values were resampled with an additional Jacobian correction to ensure preservation of the areal quantities (*Winkler et al., 2012*). We then applied an 8 mm FWHM Gaussian kernel to surface-smooth the LH and RH data.

# Data analysis

All analyses were performed in FreeSurfer (v6.0) and R (v4.1.1).

## Population-level asymmetry

We assessed areal and thickness asymmetry vertex-wise using FreeSurfer's Linear Mixed Effects (LME) tool (*Bernal-Rusiel et al., 2013*). Asymmetry was delineated via the main effect of Hemisphere (controlling for Age, Age × Hemisphere, Sex, Scanner [where applicable], with a random subject term). For each sample and metric, we computed mean Asymmetry Index maps (AI; defined as (LH-RH) / ((LH + RH)/2)). Spatial overlap of AI maps across datasets was quantified by correlating the one-dimensional surface data between every dataset pair (Pearson's r). Next, to delineate regions exhibiting robust areal and thickness asymmetry across datasets, we thresholded and binarized the

AI maps by a given absolute effect size (areal = 5%; thickness = 1%; achieving $p$[FDR]<0.001 in most datasets with FreeSurfer's two-stage FDR-procedure *Bernal-Rusiel et al., 2013*), and summed the binary maps. After removing the smallest clusters (<200 mm²), a set of robust clusters was defined as those exhibiting overlapping effects in six out of seven samples. We then extracted area and thickness data in symmetrical space for each cluster, subject, and hemisphere, spatially averaging across vertices.

## Lifespan trajectories

Factor-smooth GAMMs ('gamm4', v0.2–6 *Wood and Scheipl, 2017*) were used to fit a smooth Age trajectory per Hemisphere, and assess the smooth Age × Hemisphere interaction in our clusters. GAMMs incorporate both cross-sectional and longitudinal data to capture nonlinearity of the mean level trajectories across persons, resulting in population estimates that are intermediate between cross-sectional and longitudinal trajectories (*Sørensen et al., 2021*). The linear predictor matrix of the GAMM was used to obtain asymmetry trajectories and their confidence intervals, computed as the difference between zero-centered (i.e. demeaned) hemispheric age-trajectories. We included Hemisphere as an additional fixed effect, sex and scanner as covariates-of-no-interest, and a random subject intercept. We did not consider sex differences in lifespan asymmetry change because our elected method was not well-suited to testing three-way interactions between nonlinear smooth terms. A low number of basis dimensions for each smoothing spline was chosen to guard against overfitting (knots = 6; see *Figure 3—figure supplement 1*). LCBC outliers falling >6 SD from the trajectory of either hemisphere were detected and removed on a region-wise basis (*Supplementary file 1E-F*). To calculate relative change, we refitted lifespan GAMMs adding an ICV covariate, then scaled the LH and RH fitted lifespan trajectories by the prediction at the minimum age (i.e. ~4 years). Age at peak thickness asymmetry was estimated where the CI's of absolute hemispheric trajectories were maximally non-overlapping.

## Interregional asymmetry correlations

We assessed covariance between asymmetries, separately for areal and thickness asymmetry. All individual AI's in clusters with rightward mean asymmetry were first inversed, such that positive correlations would reflect asymmetry-asymmetry relationships regardless of the direction of mean asymmetry in the cluster (i.e. higher asymmetry in the population-direction). Then, we regressed out age, sex and scanner (where applicable) from each AI, using linear mixed models after collating the data from each sample (adding random intercepts for LCBC subjects), to ensure the correction was unaffected by differences in sample age-distribution. Separately for each dataset, we then obtained the cluster-cluster correlation matrix. At this point, two strong outliers in HCP data were detected and discarded for this and all subsequent analyses (*Figure 5—figure supplement 4*). Replication was assessed using the Mantel test ('ade4' R package v1.7–18 *Dray and Dufour, 2007*) between each dataset-pair (LCBC, UKB, HCP) across 10,000 permutations. We then post-hoc tested whether covariance between areal asymmetries was related to proximity in cortex, obtaining the average geodesic distance between all clusters along the ipsilateral surface ('SurfDist' Python package v0.15.5 *Margulies et al., 2016*), and correlating pair-wise distance with pair-wise correlation coefficient (Fisher's transformed coefficients; Spearman's correlation). To post-hoc assess whether observed covariance patterns for thickness asymmetry reflected a global effect, we ran a PCA across z-transformed AI's for all thickness clusters (pre-corrected for the same covariates). Based on the results, we computed the mean AIs across all leftward clusters, and across all rightward clusters, and tested the partial correlation between mean leftward thickness asymmetry in left-asymmetric clusters and mean rightward thickness asymmetry in right-asymmetric clusters, in each of the three cohorts.

## Heritability

We assessed heritability of areal and thickness asymmetry using both SNP- and twin-based methods, both for our set of robust clusters and cortex-wide across 500 parcels (*Schaefer et al., 2017*). For cluster analyses, significance was considered at Bonferroni-corrected p<0.05 applied separately across each metric. Cortex-wide significance was considered at $p$(FDR)<0.05 (500 tests per map). For SNP-based analyses in UKB data, the final genetic sample consisted of 31,433 UKB participants (application #32048) with imaging and quality checked genetic data. We removed subjects that were outliers

based on heterozygosity [field 22027] and missingness (>0.05), mismatched genetic and reported sex [22001], sex chromosome aneuploidies [22019], and those not in the 'white British ancestry' subset [22006] (*Bycroft et al., 2018*). At variant level, after removing SNPs with minor allele frequency <0.01, data from 784,256 autosomal SNPs were used to compute a genetic relationship matrix using GCTA (v1.93.2) (*Yang et al., 2010*). For each phenotype, we first regressed out age and sex and computed z-scores. Genome-based restricted maximum likelihood (GREML) methods as implemented in GCTA were then used to compute SNP-heritability for each AI measure, applying a kinship coefficient cut-off of .025 (excluding one individual from each pair), and controlling for genetic population structure (first 10 principal components). Bivariate GREML analysis was used to test genetic correlations between asymmetry clusters (*Yang et al., 2010*). These estimate the proportion of variance two asymmetries share due to genetic influences through pleiotropic action of genes (*van Rheenen et al., 2019*). We tested genetic relationships only for cluster-pairs where both clusters exhibited significant SNP-heritability (p<0.05; pre-corrected; 78 tests for area, 55 for thickness). Significance was assessed at *p(FDR)*<0.05. Replication of heritability results was assessed using twin-based analyses in HCP data, applying AE models in 'OpenMx' (v2.19.8) (*Neale et al., 2016*). These use observed cross-twin and cross-sibling covariance to decompose the proportion of observed phenotypic variance into additive genetic effects [A], and unique environmental effects or error [E]. Data were reformatted such that rows represented family-wise observations. As is standard, we set A to be 1 for MZ twins assumed to share 100% of their segregating genes, and 0.5 for DZ twins and siblings that share 50% on average. For each phenotype we first regressed out age and sex and computed z-scores. Statistical significance was assessed by comparing model fit to a submodel with the A parameter set to 0. To test replication of genetic correlations in HCP data, bivariate twin models were employed in OpenMx. We tested twin-based genetic correlations across the same set of cluster-pairs tested in the SNP-based analyses (*Supplementary file 1H–I*; note that while this approach permitted a replication attempt, some clusters did not exhibit significant twin-based heritability). Model significance was assessed by comparing model fit to submodels with the genetic covariance parameter set to 0, and significance was assessed at *p(FDR)*<0.05.

## Associations with cognition, sex, handedness, and ICV

Finally, we assessed relationships between asymmetry in our robust clusters and general cognitive ability, handedness, sex, and ICV. For general cognition, we used the first principal component across the following 11 core UK Biobank cognitive variables (*Fawns-Ritchie and Deary, 2020*): Mean reaction time (log transformed) [field 20023], Numeric memory [4282], Fluid reasoning [20016], Matrix completion [6373], Tower rearranging [21004], Symbol digit substitution [23324], Paired associate learning [20197], Prospective memory [20018] (recoded as 1 or 0, depending on whether the instruction was remembered on the first attempt or not), Pairs matching (log) [399], Trail making A (log) [6348], Trail making B (log) [6350]. Prior to the PCA, for participants with cognitive data, data was imputed for missing cognitive variables via the 'imputePCA' function (number of estimated components tentatively optimized using general cross validation; 'missMDA' R package v1.18 *Josse and Husson, 2016*). PC1 (explaining 39.2%; *Supplementary file 1L*) was inversed to correlate negatively with age (*r*=–0.39), ensuring higher values reflected higher cognition. As fewer participants had cognitive data relative to the other variables, for each cluster we ran one set of linear models to assess the marginal effect of cognition (PC1 as predictor; age, sex, ICV controlled; N=35,198), and one set of linear models to assess the marginal effects of Handedness, Sex, and ICV in a model including all three predictors (age controlled, N=37,569 with available handedness data). For the cognitive analysis, effects identified in the imputed dataset were checked against the confidence intervals for the effect in the subset of the data with no missing cognitive variables (N=4696). Participants who self-reported as mixed handed were not included (*Sha et al., 2021a*). Individual AI's in rightward clusters were first inversed, such that higher values reflect higher asymmetry in the population-direction. Significance was considered at Bonferroni-corrected $\alpha=p < 7.4e^{-5}$ (.01/136 [34 clusters ×4]).

## Data sharing/availability

All summary-level maps are available in *Supplementary file 2*. All code underlying the main analyses is available at https://github.com/jamesmroe/PopAsym (copy archived at *Roe, 2023*) and on the Open Science Framework (OSF; https://osf.io/dv9um/). Derived source data underlying figures is

available on the OSF. All datasets used in this work are openly available, with the exception of LCBC where participants, which include many children, have not consented to share their data publicly online. Other datasets used in this work are available without restrictions and are not subject to application approval (DLBS; https://fcon_1000.projects.nitrc.org/indi/retro/dlbs.html; CC BY-NC; SALD; http://fcon_1000.projects.nitrc.org/indi/retro/sald.html; CC BY-NC; IXI; https://brain-development.org/ixi-dataset; CC BY-SA 3.0). Accordingly, we have made the individual-level data for these samples available and our code can be used to reproduce vertex-wise analyses in these samples. Individual-level data for the remaining samples (LCBC; Cam-CAN, HCP; UKB) may be available upon reasonable request, given appropriate ethical, data protection, and data-sharing agreements where applicable. Requests must be submitted and approved via the relevant channel (details are provided in *Supplementary file 1M*).

## Acknowledgements

Scripts were run on the Colossus processing cluster at the University of Oslo, and on resources provided by UNINETT Sigma2 (nn9769k). LCBC funding: European Research Council under grants 283634, 725025 (to A.M.F.), and 313440 (to K.B.W.); Norwegian Research Council (to A.M.F. and K.B.W.) under grants 249931 (TOPPFORSK) and 302854 (FRIPRO; to Y.W.), The National Association for Public Health's dementia research program, Norway (to A.M.F). Data used in the preparation of this work were obtained from the MGH-USC Human Connectome Project (https://ida.loni.usc.edu/login.jsp). Data used in this work was also provided by the Cambridge Centre for Ageing and Neuroscience (Cam-CAN). This research has been conducted using the UK Biobank Resource (February 2020 data release). The authors also express thanks to Liyuan Yang for notifying us about the outstanding issue in HCP data.

## Additional information

### Funding

| Funder | Grant reference number | Author |
| --- | --- | --- |
| European Research Council | 283634 | Anders M Fjell |
| European Research Council | 725025 | Anders M Fjell |
| European Research Council | 313440 | Kristine B Walhovd |
| Norwegian Research Council | 249931 | Anders M Fjell<br>Kristine B Walhovd |
| Norwegian Research Council | 302854 | Yunpeng Wang |

The funders had no role in study design, data collection and interpretation, or the decision to submit the work for publication.

### Author contributions

James M Roe, Conceptualization, Data curation, Software, Formal analysis, Validation, Investigation, Visualization, Methodology, Writing - original draft, Project administration, Writing – review and editing; Didac Vidal-Pineiro, René Westerhausen, Conceptualization, Supervision, Writing – review and editing; Inge K Amlien, Mengyu Pan, Data curation; Markus H Sneve, Espen M Eilertsen, Formal analysis, Writing – review and editing; Michel Thiebaut de Schotten, Patrick Friedrich, Zhiqiang Sha, Clyde Francks, Conceptualization, Writing – review and editing; Yunpeng Wang, Conceptualization, Formal analysis, Writing – review and editing; Kristine B Walhovd, Anders M Fjell, Funding acquisition, Writing – review and editing

### Author ORCIDs

James M Roe http://orcid.org/0000-0002-8008-902X

Didac Vidal-Pineiro  http://orcid.org/0000-0001-9997-9156
Michel Thiebaut de Schotten  http://orcid.org/0000-0002-0329-1814
Clyde Francks  http://orcid.org/0000-0002-9098-890X
Kristine B Walhovd  http://orcid.org/0000-0003-1918-1123
René Westerhausen  http://orcid.org/0000-0001-7107-2712

### Ethics

All studies were conducted in accordance with the Declaration of Helsinki. Ethical approval was obtained from the relevant authorities, and all participants provided informed consent. Studies conducted at the Center for Lifespan Changes in Brain and Cognition (LCBC) were approved by the Regional Ethical Committee of South-East Norway (2017/653) and complied with all relevant ethical regulations. Ethical approval for the other datasets was granted by the relevant authorities (Supplementary file 1M).

### Decision letter and Author response

Decision letter https://doi.org/10.7554/eLife.84685.sa1
Author response https://doi.org/10.7554/eLife.84685.sa2

## Additional files

### Supplementary files

• Supplementary file 1. Supplementary tables A-M. **(A)** Demographics of the samples used for each analysis. The number of unique participants (N unique), total observations (N obs), and number of scans constituting longitudinal observations (N longitudinal) is shown. Note that only the LCBC sample included longitudinal data (70% longitudinal coverage). * See Supplementary file 1C for further details of the HCP extended twin design used for heritability analysis. ** For analyses of SNP-heritability and *** associations with individual differences, to maximize power to detect effects subsets based on maximum data availability were taken from the N=38,171 UK Biobank base sample described here (i.e., all individuals with genotype data surpassing quality control; all individuals with available cognitive/handedness data; see Methods). **(B)** MRI acquisition parameters by sample. TR = Repetition time; TE = Echo time; TI = Inversion time; FA = Flip angle; FOV = Field of view; 3D MPRAGE = three-dimensional magnetization prepared rapid gradient echo. * see available IXI imaging parameters here. **See UK Biobank brain imaging documentation here. (-) Not available/ found. **(C)** Kinships of the extended twin design used for heritability analysis (HCP data). The number of observations of each pedigree type is given as well as the overall number of subjects per pedigree-type (N; Total N=1037). As no pedigree-type contained more than a single twin-pair, the number of monozygotic (MZ) and dizygotic (DZ) twin-pairs is reported per type. As all twins were same-sex, the ratio of female/male twin pairs is given per type. **(D)** Linear regression results of the main effect of Cluster Type (i.e. Desikan-Killiany [DK] parcels v robust asymmetry clusters) upon the average correlation across vertex-mean correlations within parcels/clusters, controlling for cluster size (nVertices) and the Cluster Type × nVertices interaction. Average vertex-mean correlations were significantly higher for robust asymmetry clusters for areal asymmetry, and were significant or at trend-level for thickness asymmetry across each replication dataset. **(E)** GAMM lifespan results for age-related change in areal asymmetry in robust asymmetry clusters (population-level areal asymmetries). A smooth Age × Hemisphere interaction [s(LH-Age)-s(RH-Age)] was modelled to determine whether asymmetry exhibited significant change across the lifespan. Significance of the smooth interaction (Bonferroni corrected; $\alpha<.05/14=.0036$) is in bold. Effect size is denoted by $\Omega^2$. Hemisphere, Sex and Scanner (not shown) were additionally modelled as fixed effects covariates. Model fit is provided ($r^2$ adjusted). The number of observations is given (including both hemispheres) after removing outliers on a cluster-wise basis (defined as observations >6 SD from the fitted trajectory of either hemisphere). Edf = estimated degrees of freedom (index of curve complexity). **(F)** GAMM lifespan results for age-related change in thickness asymmetry in robust asymmetry clusters (population-level thickness asymmetries). A smooth Age × Hemisphere interaction [s(LH-Age)-s(RH-Age)] was modelled to determine whether asymmetry exhibited significant change across the lifespan. Significance of the smooth interaction (Bonferroni corrected; $\alpha<.05/20=.0025$) is in bold. Effect size is denoted by $\Omega2$. Hemisphere, Sex and Scanner (not shown) were additionally modelled as fixed effects covariates. Model fit is provided ($r^2$ adjusted). The number of observations is given (including both hemispheres) after removing outliers on a cluster-wise basis (defined as observations >6 SD from the fitted trajectory of either hemisphere). Edf = estimates degrees of

freedom (index of curve complexity). **(G)** Heritability of global brain measures. Significant twin-based (HCP data) and SNP-based (UK Biobank data; N=31,433) heritability ($h^2$) estimates are shown in bold, Bonferroni-corrected ($P<8.3e-3$ [.05/6]) separately for twin- and SNP-based estimates. Significant and high heritability estimates were observed for global brain measures of each hemisphere separately. For global asymmetry, twin-based heritability estimates were significant for both area and thickness, whereas only global areal asymmetry showed significant SNP-heritability in UK Biobank. $e^2$=unique environmental effects + error; $-2LL$ = minus 2 log likelihood index of model fit for AE (full model) and E (model without genetic parameter). Note that the correlations for DZ twins or sibling-pairs (rDZsibPairs) do not include data from third +siblings (sibs). Instead, we computed the correlation after concatenating data across all DZ twin-pairs, across twin1 (whether MZ or DZ) and sib1, or across sib1 and sib2 (where families had only siblings and no twins), all of which share 50% genetics on average. As there was always only one twin-pair in a family, the correlation for MZ twins (rMZ) is computed across all MZ twin-pair observations. **(H)** Heritability of areal asymmetry in robust asymmetry clusters (population-level areal asymmetries). Significant twin-based (HCP data) and SNP-based (UK Biobank data; N=31,433) heritability ($h^2$) estimates are shown in bold, Bonferroni-corrected ($P<3.6e-3$ [.05/14]) separately for twin- and SNP-based estimates. (*van Kesteren and Kievit, 2021*) Note the highest SNP-heritability was observed for areal asymmetry of the anterior insula ($h^2_{SNP}$ = 18.6%). Note also that because the p-value was lower than the minimum in GCTA ($P=0$) we report $P<1.78e-15$ in the main text, which was the lowest numeric p-value in cortex-wide analyses. $e^2$=unique environmental effects + error; $-2LL$ = minus 2 log likelihood index of model fit for AE (full model) and E (model without genetic parameter). Note that the correlations for DZ twins or sibling-pairs (rDZsibPairs) do not include data from third +siblings. Instead, we computed the correlation after concatenating data across all DZ twin-pairs, across twin1 (whether MZ or DZ) and sib1, or across sib1 and sib2 (where families had only siblings and no twins), all of which share 50% genetics on average. As there was always only one twin-pair in a family, the correlation for MZ twins (rMZ) is computed across all MZ twin-pair observations. **(I)** Heritability of thickness asymmetry in robust asymmetry clusters (population-level thickness asymmetries). Significant twin-based (HCP) and SNP-based (UK Biobank; N=31,433) heritability ($h^2$) estimates are shown in bold, Bonferroni-corrected ($P<2.5e-3$ [.05/20]) separately for twin- and SNP-based estimates. $e^2$=unique environmental effects + error; $-2LL$ = minus 2 log likelihood index of model fit for AE (full model) and E (model without genetic parameter). Note that the correlations for DZ twins or sibling-pairs (rDZsibPairs) do not include data from third +siblings (sibs). Instead, we computed the correlation after concatenating data across all DZ twin-pairs, across twin1 (whether MZ or DZ) and sib1, or across sib1 and sib2 (where families had only siblings and no twins), all of which share 50% genetics on average. As there was always only one twin-pair in a family, the correlation for MZ twins (rMZ) is computed across all MZ twin-pair observations. **(J)** Areal asymmetry associations. Results of linear regressions modelling the effects of individual differences (General Cognitive Ability, Handedness, Sex, and ICV) on asymmetry in population-level areal asymmetries in the full UK Biobank imaging sample with available cognitive data (imputed for no missing cognitive variables; N=35,198) or available handedness data (Handedness, Sex, ICV effects; N=37,569; Methods). Bold indicates Bonferroni-corrected significance ($P<7.4e^{-5}$ [.01/136]). Right table shows significance of associations after controlling for additional brain-size related covariates following recent recommendations (*Williams et al., 2022*). **(K)** Thickness asymmetry associations. Results of linear regressions modelling the effects of individual differences (General Cognitive Ability, Handedness, Sex, and ICV) on asymmetry in population-level thickness asymmetries in the full UK Biobank imaging sample with available cognitive data (imputed for no missing cognitive variables; N=35,198) or available handedness data (Handedness, Sex, ICV effects; N=37,569; Methods). Bold indicates Bonferroni-corrected significance ($P<7.4e^{-5}$ [.01/136]). Right table shows significance of associations after controlling for additional brain-size related covariates following recent recommendations (*Williams et al., 2022*). **(L)** Weightings of principal components across the 11 core cognitive variables in UK Biobank used here and cumulative variance explained. **(M)** Dataset access information.

- Supplementary file 2. Cortical asymmetry and heritability maps (zipped folder).
- MDAR checklist

## Data availability

All summary-level maps are available in Supplementary file 2. All code underlying the main analyses is available at https://github.com/jamesmroe/PopAsym (copy archieved at *Roe, 2023*) and on the Open Science Framework (OSF; https://osf.io/dv9um/). Derived source data underlying figures is also available on the OSF. All datasets used in this work are openly available, with the exception

of LCBC where participants, which include many children, have not consented to share their data publicly online. Other datasets used in this work are available without restrictions and are not subject to application approval (DLBS; https://fcon_1000.projects.nitrc.org/indi/retro/dlbs.html; CC BY-NC; SALD; http://fcon_1000.projects.nitrc.org/indi/retro/sald.html; CC BY-NC; IXI; https://brain-development.org/ixi-dataset; CC BY-SA 3.0). Accordingly, we have made the individual-level data for these samples available and our code can be used to reproduce vertex-wise analyses in these samples. Individual-level data for the remaining samples (LCBC; Cam-CAN, HCP; UKB) may be available upon reasonable request, given appropriate ethical, data protection, and data-sharing agreements where applicable. Requests must be submitted and approved via the relevant channel (details are provided in *Supplementary file 1*).

The following dataset was generated:

| Author(s) | Year | Dataset title | Dataset URL | Database and Identifier |
|---|---|---|---|---|
| Roe JM | 2023 | PopAsym | https://osf.io/dv9um/ | Open Science Framework, dv9um |

The following previously published datasets were used:

| Author(s) | Year | Dataset title | Dataset URL | Database and Identifier |
|---|---|---|---|---|
| Tyler LK | 2017 | Cambridge Centre for Ageing and Neuroscience (Cam-CAN) dataset inventory | https://camcan-archive.mrc-cbu.cam.ac.uk/dataaccess/ | Cambridge Centre for Ageing and Neuroscience, camcan-archive |
| Park DC | 2012 | Dallas Lifespan Brain Study (DLBS) | https://fcon_1000.projects.nitrc.org/indi/retro/dlbs.html | International Neuroimaging Data-Sharing Initiative, dlbs |
| WU-Minn Human Connectome Project (HCP) consortium | 2017 | HCP 1200 Subjects Data Release | https://www.humanconnectome.org/study/hcp-young-adult/document/1200-subjects-data-release | Connectome Coordination Facility, 1200-subjects-data-release |
| UK Biobank | 2020 | UK Biobank | https://www.ukbiobank.ac.uk/enable-your-research/apply-for-access | UK Biobank, UKB |
| Wei D, Zhuang K, Chen Q, Yang W, Liu W, Wang K, Qiu J | 2018 | Southwest University Adult Lifespan Dataset (SALD) | http://fcon_1000.projects.nitrc.org/indi/retro/sald.html | International Neuroimaging Data-Sharing Initiative, sald |
| Biomedical Image Analysis Group, Imperial College London, Centre for the Developing Brain, King's College London | 2023 | IXI Dataset | https://brain-development.org/ixi-dataset/ | brain-development.org, IXI |

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
