## [Editor Report]

Roe et al. provide a large-sample analysis of hemispheric lateralisation in brain structure, synthesising local cortical thickness and surface area data from 7 different datasets. The study provides a rich descriptive catalogue of phenomena related to hemispheric anatomical asymmetries. These results are convincing and will prove an important point of reference to neuroscientists who might want to compare their own future results to the ones from this large and varied data set.

---

## [Decision Letter]

**Decision letter after peer review:**

[Editors’ note: the authors submitted for reconsideration following the decision after peer review. What follows is the decision letter after the first round of review.]

Thank you for submitting the paper "Population-level asymmetry of the cerebral cortex: reproducibility, lifespan changes, heritability, and individual differences" for consideration by *eLife*. Your article has been reviewed by 3 peer reviewers, and the evaluation has been overseen by a Reviewing Editor and a Senior Editor. The following individual involved in the review of your submission has agreed to reveal their identity: Dorothy V M Bishop (Reviewer #3).

Comments to the Authors:

We are sorry to say that, after consultation with the reviewers, we have decided that this work in its current form will not be considered further for publication by *eLife*.

However, note that in the light of the very constructive comment and in tune with our general belief that these data can potentially provide an important reference for future studies, we as editors would be willing to consider a future, thoroughly revised revision of this paper. We would treat it as a new submission (with potentially new reviewers being invited, depending on the availability of reviewers.) – All of this of course depending on the nature and extent of your revisions to the manuscript and your documentation of these. *

Specifically, the reviewers and editors agreed that the paper provides a useful descriptive catalogue of phenomena related to hemispheric anatomical asymmetries and that it does so in a largely sound manner (but see specific methodological concerns below). However, there was no consensus to be reached that this study has a specific impact on our learning about the mechanisms of hemispheric lateralisation. After reviews and an elaborate discussion amongst reviewers and editors, concerns remained over the mechanistic impact of the results presented and the degree of conceptual synthesis provided, so we decided not to invite a revision of this study for *eLife*.

*Reviewer #1 (Recommendations for the authors):*

I have several comments and suggestions that I hope will be of use to the authors.

1. In discussing the differences between regional asymmetries in cortical thickness and cortical area, it is prudent to mention that CT is less reliable than SA and this reliability difference may be amplified in the computation of asymmetry.

2. In claiming that "disrupted cortical asymmetry is a confirmed feature of… aging…" the authors cite one study and ignore others that not only lend little support to this conclusion but also warn that regional asymmetry in the aging brain is not well-replicated even across similar samples.

3. Unclear how GAMM captures nonlinearity of change over time with only two measurement occasions.

4. Discovering a capture of 21% of common variance by the first component of the PCA hardly signals the existence of a single factor. CFA would be a more appropriate way of testing single vs multiple factors hypotheses.

5. The authors consistently characterize correlations as "strong" – unclear why. The characterization is somewhat hyperbolic: for example, r=.56 does not seem particularly strong, with about 70% of the variance not shared by the variables. Why not just write "correlated," report the magnitude and the direction and let the readers make a qualitative judgment if they feel like it.

Some stylistic editing will improve this manuscript. For example, the sentence "Areal asymmetry correlates in specific regions…" in the Abstract is quite unwieldy and can be split into two or three.

*Reviewer #2 (Recommendations for the authors):*

I found some of the graphs hard to read.

For example, in Figure 5 the color scale was hard to read. It goes from one shade of blue to another shade of blue and I could not really see much distinction.

Figure 6, it is not clear what is being plotted on the abscissa. There is no axis label.

The authors use the term "longitudinal" to describe their age-related analyses, but my understanding of that term is that it refers to the same individuals tested over time, which is not the case here; it's a cross-sectional approach.

The word "course" should be "coarse" (line 70).

*Reviewer #3 (Recommendations for the authors):*

Overall, while I think the attempt at a comprehensive approach by the authors is commendable, the paper, which includes a great deal of supplementary material, is just too dense and unwieldy. My recommendation would be to break it up. I'm not a fan of authors who turn papers into 'minimal publishable units', but here we have the opposite – a paper so dense with information that it is hard to follow.

I don't know if *eLife* would consider a series of related papers like this but the work would be easier to follow, and potentially more influential if it were split up. There is a nice logical progression, with the initial analyses delineating brain areas that are consistently asymmetric, and subsequent analyses then using these, so a series of papers would work well I think – something like this:

Paper 1: qs 1 and 3 – replicability of population-level asymmetries, and inter-regional correlations.

Paper 2: consideration of causal factors, incorporating longitudinal analysis of lifespan trajectories of asymmetry (q2) and heritability (q4). The longitudinal analysis could potentially incorporate analysis of interregional correlations across age – given that connectivity changes with age, there is a rationale for looking at that (LCBC sample), but even without that, this would make a meaty paper.

Paper 3 (q5) Consideration of significance of individual differences in terms of correlations with sex, handedness, and general cognitive ability (UKB sample). Potentially more could be done with this; it could look at associations with absolute as well as raw asymmetry indices. But I think it would be most valuable if preceded by a balanced review of prior work in this area, culminating in a list of claims of associations that could be tested in this sample. That would make it more evident which claims are NOT supported, which would be very useful for the field.

Note: I am not recommending jettisoning the null results – on the contrary, I would like to see them given more serious treatment, as they could be helpful in moving the field forward.

Since breaking up the paper this way would be a fairly radical move, I'll assume that the authors and/or editors will decide against this, and so my remaining comments will be written as if we are dealing with a paper that does incorporate all the current material.

I would suggest rewriting the Abstract to give a more balanced account of the findings, including an indication of the null as well as positive results.

I don't know whether *eLife* insists on the particular structuring of the paper used here, with Methods at the end. I would hope this could be modified, as for a multi-question paper like this, it is particularly frustrating for the reader. The introduction ends with five specific questions clearly laid out, but we then jump into Results, which are hard to understand without seeing the Methods. So the reader goes forward to Methods, where the dataset for each of the five questions is first described, then the data analysis methods for each of the five questions. This involves a great deal of jumping about. I found myself wanting to literally cut and paste the material so I could read it in a sensible sequence.

Interregional correlations

The material on interregional correlations was fascinating but hard to follow. The numbers (Figure 4 Figure supplement 1) are hard to read, but I was very surprised at just how low these correlations were for SA, given that people have tended to interpret asymmetries in terms of a general 'torque' effect.

The CT asymmetries were hard to interpret, given the relatively low Mantel scores – and the way the Biobank data stand out even just on the heatmap. This made me wonder if the more restricted age range of the Biobank sample might affect the covariances. I wasn't sure how the age residualisation worked, and whether it was within-sample or across samples, so hard to know if and how the fact that Biobank has a much higher age minimum than others might have an effect?

One thing that makes the results confusing is the flipping of the hemisphere so that the emphasis is on strength of laterality in the population direction. It would be worth taking some time in the introduction to discuss predictions from the 'multifactorial' account versus a 'single factor' account – relating this to discussions of cerebral torque.

An alternative approach would be to just present the unflipped data, explaining that a positive correlation then indicates agreement in left-wardness (regardless of whether the population bias is left or right), and a negative correlation indicates agreement between the strength of lateralisation in the population direction.

The subsequent discussion of these results was pretty clear – it's just the account in Results that was hard to follow.

Heritability

The account of results in the Abstract again focused on the positive findings and ignored the null results as if they were of no interest.

So the Abstract states: "Areal asymmetry is moderately heritable (max h2SNP ~19%), and phenotypic correlations are reflected by high genetic correlations, whereas heritability of thickness asymmetry is low. "

My reading of the results is that none of the twin-based heritability estimates – either global or regional – was statistically significant. Consistent with previous reports by Eyler et al. and others, the strongest influence on asymmetry indices is the E-term (nonshared environment), which may be interpreted in this context as chance (stochastic) influences.

The SNP-based estimates from UK Biobank give a handful of significant heritability estimates for the SA asymmetry, but none for CT. I agree it is of interest to document these, but the genetic effects that are found are not high (19% is the highest estimate, with the remaining 80% of variance accounted for by the e2 term).

I guess it is a case of one's perspective, but my take on this is not "isn't it interesting that if we have a huge sample we can pick up some small genetic effects on a handful of asymmetry measures", but rather, "isn't it interesting that brains end up with similar patterns of structural asymmetry, even though genes explain so little individual variation ". This does suggest that there are genetic influences that create these general biases to one direction or the other, but which have reached fixation in the population and so show little individual variation.

Associations with cognition

As noted above, I felt this treatment was too brief. There is a rich literature on associations with asymmetry for the variables considered here, much of it focusing on the degree rather than the direction of asymmetry. So I would like to see a separate paper dealing with associations with AIs from this analysis, relating the findings more systematically to prior studies.

The discussion did a phenomenal job in pulling everything together but would benefit from some subheadings.

[Editors’ note: further revisions were suggested prior to acceptance, as described below.]

Thank you for submitting your article "Tracing the development and lifespan change of population-level structural asymmetry in the cerebral cortex" for consideration by *eLife*. Your article has been reviewed by 1 (new) peer reviewer who had also access to your rebuttal letter. The evaluation has been overseen by Jonas Obleser as the Reviewing editor and Christian Büchel as the Senior Editor. The reviewer has opted to remain anonymous. The Reviewing Editor has drafted this to help you prepare a revised submission.

Essential revisions (for the authors):

1) Thorough editions for clarity. This new reviewer has overall praise for your work, but they also expose the remaining lack of clarity and a relative lack of depth in coverage of the genetic results. Please use the detailed comments below as a guide when revising your study prior to final acceptance.

2) Reproducibility: In addition to revising along the reviewer's comments, …

– Please also fix issues such as "All maps are available at neurovault.org/XXXX."

­

­– Please add version numbers where applicable (eg to R packages).

*Reviewer #1 (Recommendations for the authors):*

Overall I believe the authors have done a solid job in assessing brain asymmetry and addressing the comments of the reviewers.

Nevertheless, here and there some elements may still be (considered to be) tidied up a bit further, so that the aim of the study is even more clear. Currently, there are still a lot of sub-analyses. For example, 2.3 is interesting but possibly a slight deviation from the general goal.

Moreover, with respect to the analysis part 2.5, did the authors consider running these analyses also in HCP?

Also – with respect to the heritability analyses; have genetic correlations also been performed in the HCP sample?

---

## [Author Response]

[Editors’ note: the authors resubmitted a revised version of the paper for consideration. What follows is the authors’ response to the first round of review.]

Reviewer #1 (Recommendations for the authors):I have several comments and suggestions that I hope will be of use to the authors.1. In discussing the differences between regional asymmetries in cortical thickness and cortical area, it is prudent to mention that CT is less reliable than SA and this reliability difference may be amplified in the computation of asymmetry.

We agree and have added text to reflect this in the Discussion (see below). Furthermore, we now also acknowledge that this potential reliability difference may be a contributor to some of the differences in results for areal and thickness asymmetry (e.g. heritability), stating this in the limitations:

615 – “Conceivable sources of inconsistently reported results for cortical asymmetries likely include varying age distributions (particularly for thickness) (Roe et al. 2021), and multiple asymmetries within atlas-based parcels (e.g. we observed notably discrepant results to ENIGMA for insula thickness asymmetry (Kong et al. 2018)) … Thickness asymmetries seem also more variable compared with area (Kong et al. 2018), though it should be noted thickness in general may be slightly less reliable (Hedges et al. 2022), the asymmetry effect is smaller, and thus possibly contains more error”

646 – “Again, we also cannot rule out that potential reliability differences between thickness and areal asymmetry may partially explain some magnitude differences in heritability estimates.”

2. In claiming that "disrupted cortical asymmetry is a confirmed feature of… aging…" the authors cite one study and ignore others that not only lend little support to this conclusion but also warn that regional asymmetry in the aging brain is not well-replicated even across similar samples.

We apologize for the lack of clarity. In the context of aging, this statement is specific to cortical thickness asymmetry. While we agree previous cross-sectional studies provided mixed support for this (e.g. (Zhou et al. 2013; Plessen et al. 2014)), our previous study which we cite here showed that both leftward and rightward thickness asymmetry exhibit widespread reduction in aging that is replicable in multiple longitudinal samples (Roe et al. 2021). This previous study was the first to address the question longitudinally, which is likely the key reason why earlier cross-sectional studies lent little support to this conclusion. We agree the few studies assessing cross-sectional age effects on areal asymmetries do not lend support for the claim in aging, and indeed, a key motivation for the current study is that we know little about lifespan changes in areal asymmetry. As this is an introductory sentence, it is perhaps too early to specify that in the context of neurodevelopmental disorders, the evidence suggests alterations in both area and thickness, whereas in the context of aging, current evidence supports changes in thickness only. Therefore, we opted for the following change that highlights only some aspects of cortical asymmetry have been shown to be subtly altered in aging, without going into the specifics at this point – which we do later in paragraphs dealing with these issues in depth. We agree reported patterns of average regional asymmetry in the aging brain in general have been inconsistent, especially for thickness asymmetry (e.g. Zhou et al. 2013; Plessen et al. 2014; Roe et al. 2021). We believe this point only highlights the need for our study and approach.

55 – ”Although an extensive literature in search of structural asymmetry deviations in various conditions and disorders is in several cases being challenged by newer data (Kong et al. 2020), at least some aspects of cortical asymmetry are confirmed to be subtly reduced in neurodevelopmental disorders such as autism (Postema et al. 2019), but also through later life influences such as aging (Roe et al. 2021), and Alzheimer’s disease (Thompson et al. 2007; Roe et al. 2021)”

74 – “Determining the developmental and lifespan trajectories of cortical asymmetry may shed light on how cortical asymmetries are shaped through childhood or set from early life, and provide evidence of the timing of expected brain change in normal development. Although important in and of itself, this would also provide a useful normative reference, as subtly altered cortical asymmetry – in terms of both area and thickness – has been linked at the group-level to neurodevelopmental disorders along the autism spectrum (Postema et al. 2019; Sha et al. 2022), suggesting altered lateralized neurodevelopment may be a neurobiologically relevant outcome in at least some cases of developmental perturbation (Postema et al. 2019; Sha et al. 2022). For areal asymmetry, surprisingly few studies have charted developmental (Li et al. 2014; Remer et al. 2017) or aging-related effects (Kong et al. 2018; Williams et al. 2022), although indirect evidence in neonates suggests adult-like patterns of areal asymmetry are evident at birth and may exhibit little change from birth to 2 years (Li et al. 2014) despite rapid and concurrent developmental cortical expansion (Li et al. 2013). For thickness asymmetry, longitudinal increases in asymmetry have been shown during the first two years of life (Li et al. 2015), with suggestions of rapid asymmetry growth from birth to 1 year (Li et al. 2015), and potentially continued growth until adolescence (Nie et al. 2013). However, previous lifespan studies mapped thickness asymmetry linearly across cross-sectional developmental and adult age-ranges (Zhou et al. 2013; Plessen et al. 2014), mostly concluding thickness asymmetry is minimal in infancy and maximal age ~60. In contrast, recent work established thickness asymmetry shows a non-linear decline from 20 to 90 years that is reproducible across longitudinal aging cohorts (Roe et al. 2021). Thus, although offering viable developmental insights (Zhou et al. 2013; Plessen et al. 2014), previous lifespan studies of thickness asymmetry do not accurately capture the aging process, and likely conflate non-linear developmental and aging trajectories with linear models. A longitudinal exploration of the lifespan trajectories of thickness asymmetry accounting for dynamic change is needed to further knowledge of normal human brain development.”

3. Unclear how GAMM captures nonlinearity of change over time with only two measurement occasions.

We agree this is useful to clarify. GAMMs capture the non-linearity of the mean-level trajectories across persons, incorporating data from both cross-sectional and longitudinal measurements. They do not model non-linearity within persons. Hence, the non-linearity in GAMM is achieved in the same way as in a Generalized Additive Model that does not incorporate repeat measures (essentially a weighted sum of basis functions and their coefficients). Since both single and repeat measures over time are taken into account, the estimated population trajectories will be an intermediate estimate of cross-sectional and longitudinal trajectories. We have added this clarification in the Methods:

743 – “GAMMs incorporate both cross-sectional and longitudinal data to capture non-linearity of the mean level trajectories across persons (Sørensen et al. 2021).”

4. Discovering a capture of 21% of common variance by the first component of the PCA hardly signals the existence of a single factor. CFA would be a more appropriate way of testing single vs multiple factors hypotheses.

We have changed all statements pertaining to a “single global factor” to a “single principle component” or similar, as we agree this better represents our findings.

5. The authors consistently characterize correlations as "strong" – unclear why. The characterization is somewhat hyperbolic: for example, r=.56 does not seem particularly strong, with about 70% of the variance not shared by the variables. Why not just write "correlated," report the magnitude and the direction and let the readers make a qualitative judgment if they feel like it.

We have removed such examples where this refers to a correlation. We now only use the word “strong” when describing asymmetries.

Some stylistic editing will improve this manuscript. For example, the sentence "Areal asymmetry correlates in specific regions…" in the Abstract is quite unwieldy and can be split into two or three.

In line with this suggestion and others, we have made extensive amendments to the abstract. However, since we must be mindful of word count and taking the care to highlight our main findings, we cannot split this sentence into multiple sentences. The abstract now reads:

Abstract:

31 – “Cortical asymmetry is a ubiquitous feature of brain organization that is subtly altered in some neurodevelopmental disorders, yet we lack knowledge of how its development proceeds across life in health. Achieving consensus on the precise cortical asymmetries in humans is necessary to uncover the genetic and later influences that shape them, such as age. Here, we delineate population-level asymmetry in cortical thickness and surface area vertex-wise in 7 datasets and chart asymmetry trajectories longitudinally across life (4-89 years; observations = 3937; 70% longitudinal). We find replicable asymmetry interrelationships, heritability maps, and test asymmetry associations in large-scale data. Cortical asymmetry was robust across datasets. Whereas areal asymmetry is predominantly stable across life, thickness asymmetry grows in childhood and peaks in early adulthood. Areal asymmetry correlates phenotypically and genetically in specific regions, and is low-moderately heritable (max h^2^_SNP_ ~19%). In contrast, thickness asymmetry is globally interrelated across the cortex in a pattern suggesting highly left-lateralized individuals tend towards left-lateralization also in population-level right-asymmetric regions (and vice versa), and exhibits low or absent heritability. We find less areal asymmetry in the most consistently lateralized region in humans associates with subtly lower cognitive ability, and confirm small handedness and sex effects. Results suggest areal asymmetry is developmentally stable and arises in early life through genetic but mainly subject-specific stochastic effects, whereas childhood developmental growth shapes thickness asymmetry and may lead to directional variability of global thickness lateralization in the population.”

Reviewer #2 (Recommendations for the authors):I found some of the graphs hard to read.For example, in Figure 5 the color scale was hard to read. It goes from one shade of blue to another shade of blue and I could not really see much distinction.

We have changed the scale in Figure 5 to a warm colour scale, and changed the gradient to better distinguish between numerical estimates. Since we have also added a new figure 4, please see revised Figure 6.

Figure 6, it is not clear what is being plotted on the abscissa. There is no axis label.

We have added a label that makes clear the x-axis plots the individual association tests per cluster (“Cluster associations”). That is, for areal asymmetry which has 14 clusters, 14 associations are conducted for general cognitive ability, handedness etc. Please see revised Figure 7.

The authors use the term "longitudinal" to describe their age-related analyses, but my understanding of that term is that it refers to the same individuals tested over time, which is not the case here; it's a cross-sectional approach.

As stated above, we apologize this very important point was not clear, but the LCBC sample used for the lifespan trajectory analysis did indeed incorporate dense longitudinal data (70% of scans were longitudinal in nature, with 1-6 timepoints per person). In fact, all of the analyses ran with the LCBC sample incorporated all longitudinal data within the given age-range. Therefore, our use of this term, as well as other longitudinal terms – such as “aging” and “change” – is justified and correct. We have extensively amended the revised manuscript to highlight the longitudinal lifespan aspect to these data, which we propose was key to uncovering the developmental trajectories we show here. Some example amendments/statements:

35 – “… and chart asymmetry trajectories longitudinally across life (4-89 years; observations = 3937; 70% longitudinal)”.

61 – “no previous study has charted cortical asymmetry trajectories from childhood to old age using longitudinal data”.

63 – “Compounding the lack of longitudinal investigation, previous large-scale studies do not …”.

90 – “A longitudinal exploration of the lifespan trajectories of thickness asymmetry accounting for dynamic change is needed to further knowledge of normal human brain development.”

121 – “To gain insight into their development, we then trace a series of lifespan and genetic analyses. Specifically, we chart the developmental and lifespan trajectories of cortical asymmetry for the first time longitudinally across the lifespan.”

179 – “we aimed to characterize the developmental and lifespan trajectories of cortical asymmetry from early childhood to old age, using a lifespan sample incorporating dense longitudinal data (Methods). For this, we used the mixed effects LCBC lifespan sample covering the full age-range (4-89 years)”.

416 – “Combining the strengths of a vertex-wise delineation of population-level cortical asymmetry in 7 international datasets and dense longitudinal data, we offer the first description of the longitudinal developmental and lifespan trajectories of cortical asymmetry, advancing knowledge on normal human brain development.”

The word "course" should be "coarse" (line 70).

Fixed.

Reviewer #3 (Recommendations for the authors):Overall, while I think the attempt at a comprehensive approach by the authors is commendable, the paper, which includes a great deal of supplementary material, is just too dense and unwieldy. My recommendation would be to break it up. I'm not a fan of authors who turn papers into 'minimal publishable units', but here we have the opposite – a paper so dense with information that it is hard to follow.1. I don't know if eLife would consider a series of related papers like this but the work would be easier to follow, and potentially more influential if it were split up. There is a nice logical progression, with the initial analyses delineating brain areas that are consistently asymmetric, and subsequent analyses then using these, so a series of papers would work well I think – something like this:Paper 1: qs 1 and 3 – replicability of population-level asymmetries, and inter-regional correlations.Paper 2: consideration of causal factors, incorporating longitudinal analysis of lifespan trajectories of asymmetry (q2) and heritability (q4). The longitudinal analysis could potentially incorporate analysis of interregional correlations across age – given that connectivity changes with age, there is a rationale for looking at that (LCBC sample), but even without that, this would make a meaty paper.Paper 3 (q5) Consideration of significance of individual differences in terms of correlations with sex, handedness, and general cognitive ability (UKB sample). Potentially more could be done with this; it could look at associations with absolute as well as raw asymmetry indices. But I think it would be most valuable if preceded by a balanced review of prior work in this area, culminating in a list of claims of associations that could be tested in this sample. That would make it more evident which claims are NOT supported, which would be very useful for the field.Note: I am not recommending jettisoning the null results – on the contrary, I would like to see them given more serious treatment, as they could be helpful in moving the field forward.2. Since breaking up the paper this way would be a fairly radical move, I'll assume that the authors and/or editors will decide against this, and so my remaining comments will be written as if we are dealing with a paper that does incorporate all the current material.

Thank you for your thoughtful comments. We do not feel the work will be as impactful if split up over several papers in the current publishing landscape, and so respectfully maintain our decision not to. However, we have taken on board the Reviewer’s many comments for how to improve the paper as it is.

(Regarding the specific point about association analysis being a paper on its own, please see below):

We agree the literature review regarding asymmetry-cognition associations lacked some necessary detail. However, although a rich literature has attempted to associate diverse aspects of brain asymmetry with cognitive ability, almost none of these studies focuses specifically on cortical asymmetry. Indeed, we are only aware of three other studies testing cognitive links with cortical asymmetry or derivations of it (beyond our previous one which included cognitive change analysis in aging; (Roe et al. 2021)). Two of these focused on fluctuating asymmetry (Yeo et al. 2016; Moodie et al. 2020), operationalizing this as the average across absolute asymmetry deviance scores from the mean per-ROI, and are therefore not directly comparable to the present study. The remaining one found that greater thickness asymmetry in the expected direction may relate to better cognitive outcomes (Plessen et al. 2014). We now explicitly mention this study in the Introduction and Discussion, as well as summarize all previous work attempting to relate cortical asymmetry specifically to variations in cognitive ability:

106 – “Finally, altered development of cerebral lateralization in general has been widely hypothesized to relate to average poorer cognitive outcomes (Crow et al. 1998; Hirnstein et al. 2010; Plessen et al. 2014). Specifically in the context of cortical asymmetry, however, although one previous study reported larger thickness asymmetry may relate to better verbal and visuospatial cognition (Plessen et al. 2014), phenotypic asymmetry-cognition associations have been rarely reported (Plessen et al. 2014; Yeo et al. 2016; Moodie et al. 2020), conflicting (Yeo et al. 2016; Moodie et al. 2020), not directly comparable (Plessen et al. 2014; Moodie et al. 2020), and to date remain untested in large-scale data. Still, recent work points to small but significant overlap between genes underlying multivariate brain asymmetries and those influencing educational attainment and specific developmental disorders impacting cognition (Sha, Schijven, et al. 2021), indicating either pleiotropy between non-related traits or capturing shared genetic susceptibility to altered brain lateralization and cognitive outcomes.”

574 – “Also of note, no other asymmetry showed evidence of cognitive association (Figure 7—figure supplement 1), suggesting previously reported associations in small samples will likely not replicate (Plessen et al. 2014)”

Since we are careful not to generalize our findings to asymmetry measures not studied here, we do not agree we should separate this into another paper. Methodologically, we maintain our decision to reduce the 11 UK Biobank cognitive variables to their shared variation (Lyall et al. 2016), because the brief nature of UKB cognitive tests does not permit an overly thorough investigation into how different aspects of cognition relate to cortical asymmetry. That is, although it seemed like a brief treatment, it is a defensible and possibly close to optimal approach to take with this data. Note that we now consistently refer to our PC1 measure in terms of “general cognitive ability”. Regarding the distinction with absolute asymmetry, the statements we opt for are always specific to cortical asymmetry, and we are not aware of a large literature looking at absolute strength of cortical asymmetries when testing associations with cognition. However, we have added text to acknowledge this remains to be tested:

657 – “Relatedly, because UKB cognitive data is not exhaustive (Fawns-Ritchie and Deary 2020) (e.g. fluid IQ ranges from 1-13), we extracted the common variance across core tests to index general cognitive ability. This approach does not permit testing associations with well-operationalized or specific cognitive domains (for which UKB cognitive data may not be sufficient), and it remains to be seen whether cortical asymmetry may be informative in the context of specific forms of lateralized cognition (Ocklenburg et al. 2014), or using absolute non-directional measures. However, the subtlety of the only effect on cognition we find here may suggest we might expect similarly weak effects, though not necessarily if brain and phenotypic measurement accuracy is jointly optimized (Aki Nikolaidis et al. 2022).”

3. I would suggest rewriting the Abstract to give a more balanced account of the findings, including an indication of the null as well as positive results.

As per a previous comment by the Reviewer, we now highlight that all effects we find are small in the Abstract and Discussion, including heritability findings.

Though we acknowledge we must be careful not to interpret the null, we agree that in data of this size, we can tentatively suggest that the null relationships we find suggest any effect of a given cortical asymmetry on e.g. general cognitive ability is arguably negligible if we are not able to detect it here. We also regret including the last sentence the reviewer mentions, as we completely agree it is very unlikely measures of specific cognitive abilities will be highly informative in terms of association with cortical asymmetry. We have added this argumentation and amended the closing statement accordingly:

658 – “This approach does not permit testing associations with well-operationalized or specific cognitive domains (for which UKB cognitive data may not be sufficient), and it remains to be seen whether cortical asymmetry may be informative in the context of specific forms of lateralized cognition (Ocklenburg et al. 2014), or using absolute nondirectional measures. However, the subtlety of the only effect on cognition we find here may suggest we might expect similarly weak effects, though not necessarily if brain and phenotypic measurement accuracy is jointly optimized (Aki Nikolaidis et al. 2022)”

Though we agree with the reviewer’s points, the Abstract is arguably not the right place to present the argument for null associations, also given word limits, which also influence bias towards positive results. However, we have made changes to the Abstract to highlight the found effects were small, including heritability effects (note that in line with other Reviewer suggestions, we have reduced emphasis on the individual differences analysis and thus no longer highlight these findings to the same degree in the Abstract). In accordance with the Reviewer’s later comments, we also highlight the interpretation of stochastic influences, further highlighting that (heritability) effects were predominantly small.

38 – “Areal asymmetry correlates phenotypically and genetically in specific regions, and is low-moderately heritable (max h^2^_SNP_ ~19%). In contrast, thickness asymmetry is globally interrelated …, and exhibits low or absent heritability. We find less areal asymmetry in the most consistently lateralized region in humans associates with subtly lower cognitive ability, and confirm small handedness and sex effects. Results suggest areal asymmetry is developmentally stable and arises in early life through genetic but mainly subject-specific stochastic effects, whereas childhood developmental growth shapes thickness asymmetry and may lead to directional variability of global thickness lateralization in the population.”

We also added Discussion around null/small effects in the context of the small cognitive association we find:

572 – “However, as seems typical for brain associations in big data (Marek et al. 2022) and may be expected for any single structural measure explaining a complex phenomenon, the association was notably small. Also of note, no other asymmetry showed evidence of cognitive association (Figure 7—figure supplement 1), suggesting previously reported associations in small samples will likely not replicate (Plessen et al. 2014).”

And amended the Results and Discussion to highlight that the found associations are indeed small:

381 – “Notably, all effect sizes were small … Although small, we note this association was far from only just surviving correction at our predefined α level (⍺ = .01; Methods).”

422 – “… and uncover novel and confirm previously-reported associations with factors purportedly important in the context of asymmetry – all with small effects.”

576 – That the association we find was specific to the most lateralized areal asymmetry in humans may suggest disruptions in early life cerebral lateralization leads to cognitive deficits – detectable in later life as small effects in big data.

590 – “the small effects highlight cortical asymmetry cannot predict individual hand preference.”

591 – “Similarly, asymmetry-relationships with other factors typically assumed important were all small, despite our optimization of asymmetry phenotypes”

4. I don't know whether eLife insists on the particular structuring of the paper used here, with Methods at the end. I would hope this could be modified, as for a multi-question paper like this, it is particularly frustrating for the reader. The introduction ends with five specific questions clearly laid out, but we then jump into Results, which are hard to understand without seeing the Methods. So the reader goes forward to Methods, where the dataset for each of the five questions is first described, then the data analysis methods for each of the five questions. This involves a great deal of jumping about. I found myself wanting to literally cut and paste the material so I could read it in a sensible sequence.

We enquired, but *eLife* mostly prefer to stick to a methods last format. They did say they sometimes allow a change in structure “where it makes sense to do so”, but the general impression was that it would not be deemed justifiable. Therefore, to avoid futile work, we chose to refine the new submission to hopefully be easier to follow, in the format they predominantly accept. While we sympathize with the Reviewer that it is not always preferable, we hope for understanding in this matter, as the decision does not lie with us.

Interregional correlations5. The material on interregional correlations was fascinating but hard to follow. The numbers (Figure 4 Figure supplement 1) are hard to read, but I was very surprised at just how low these correlations were for SA, given that people have tended to interpret asymmetries in terms of a general 'torque' effect.

We reduced the number of technical analyses in this section (i.e. geodesic distance) and streamlined the results to be more readable. We have also increased the resolution of the figure (now Figure 5—figure supplement 1). We agree many of the correlations are indeed surprisingly low for areal asymmetry, and highlight multifactorial explanations in the Discussion.

510 – “For areal asymmetry, we uncovered a covariance structure that almost perfectly replicated across datasets. In general, this fit with a multifaceted view (Rentería 2012; Francks 2015; Bain et al. 2019), in which most asymmetries were either not or only weakly correlated – but reliably so – contrasting views emphasizing a single (Annett 1964, 1998) or overall anatomical factor (Crow 2010) controlling cerebral lateralization. However, we also identified several regions wherein areal asymmetry reliably correlated within individuals, showing the variance in cortical asymmetries is not always dissociable, as often thought (Rentería 2012; Francks 2015; Bain et al. 2019). The strongest relationships all pertained to asymmetries that were proximal in cortex but opposite in direction. Several of these were underpinned by high asymmetry-asymmetry genetic correlations, illustrating cerebral lateralizations in surface area that are formed under common genetic influence, and in agreement with likely prenatal origins for areal asymmetry (Rakic 1995; Li et al. 2014).”

6. The CT asymmetries were hard to interpret, given the relatively low Mantel scores – and the way the Biobank data stand out even just on the heatmap. This made me wonder if the more restricted age range of the Biobank sample might affect the covariances. I wasn't sure how the age residualisation worked, and whether it was within-sample or across samples, so hard to know if and how the fact that Biobank has a much higher age minimum than others might have an effect?

The age residualisation was previously performed separately within each sample, so it was plausible differences in age-ranges could have affected the results. Since we agree a better approach is to perform the age/sex/site/residualisation across collated samples, we followed the suggestion. The overall pattern of results did not change, with the UKB data still standing out. However, this approach improved results of the Mantel tests, increasing the whole-matrix correlation from r = .43 to.49 for the LCBC-UKB comparison, and from r = .45 to.62 for the LCBC-HCP comparison. This revised approach supports a stronger replication, with the effects clearly similar across samples, albeit still weaker in LCBC and HCP.

We have also added the following in the Discussion to mention the differences between datasets for thickness asymmetries:

526 – “Though it is unclear why the relationships were weaker in the other datasets tested, we nevertheless found similarly significant relationships in each (Figure 5—figure supplement 3).”

7. One thing that makes the results confusing is the flipping of the hemisphere so that the emphasis is on strength of laterality in the population direction.

Please see our response in point 9 below.

8. It would be worth taking some time in the introduction to discuss predictions from the 'multifactorial' account versus a 'single factor' account – relating this to discussions of cerebral torque.

We have added the following in the introduction to briefly mention these contrasting views:

93 – “Correlations between cortical asymmetries may provide a window on asymmetries formed under common genetic or developmental influences. Contemporary research suggests brain asymmetries are complex, multifactorial and largely independent (i.e. uncorrelated) traits (Rentería 2012; Francks 2015; Neubauer et al. 2020), contrasting earlier theories emphasizing a single (Annett 1964, 1998) or predominating factor (Geschwind and Galaburda 1985; McManus and Bryden 1991) controlling various cerebral lateralizations”

In addition, we also briefly mention the overall cerebral torque factor in the Discussion:

510 – “For areal asymmetry, we uncovered a covariance structure that almost perfectly replicated across datasets. In general, this fit with a multifaceted view (Rentería 2012; Francks 2015; Bain et al. 2019), in which most asymmetries were either not or only weakly correlated – but reliably so – contrasting views emphasizing a single biological (Annett 1964, 1998) or overall anatomical torque factor (Crow 2010) controlling cerebral lateralization.”

9. An alternative approach would be to just present the unflipped data, explaining that a positive correlation then indicates agreement in left-wardness (regardless of whether the population bias is left or right), and a negative correlation indicates agreement between the strength of lateralisation in the population direction.10. The subsequent discussion of these results was pretty clear – it's just the account in Results that was hard to follow.

Although we of course agree this is a viable alternative approach, we argue it is a less obvious interpretation for the average reader that a positive correlation between cluster AI’s with a negative mean (i.e. rightward) and one with a positive mean (i.e. leftward) indicates agreement in leftward asymmetry. At least, this will likely be less obvious to those outside of the laterality community, such as developmental scholars, and readers who skim the explanation in the figure caption that the “opposite hemisphere” relationships are evidence of an arguably different phenomenon may easily miss this point (see below heat maps). Instead, we chose to highlight the qualitatively different nature of the relationships we see between opposite-direction asymmetries. We argue that the most unambiguous and clear way to present these relationships is that a positive correlation should indicate an asymmetry-asymmetry relationship, regardless of population-direction in the cluster (i.e. higher asymmetry in the population-direction, as the Reviewer points out). As is made clear in the below heat maps, the flipping of course only affects the opposite-direction asymmetry relationships. The negative correlations immediately stand out as a different phenomenon, and indicate that more leftward asymmetry relates to less rightward asymmetry, without having to consider the AI formula. We chose to consistently do this also to aid interpretation of genetic correlations (again to asymmetry-asymmetry genetic relationships), and for the analysis of cognitive ability, handedness, sex etc. For the latter, it greatly facilitates the communication of results to the average reader to not have to consider that e.g. the interpretation of the relationship between handedness and asymmetry changes depending on whether the brain cluster visualized in Figure 7 and Author response image 1 is yellow or blue. For these reasons, we respectfully opt to maintain this decision. Where relevant, we have added the following text or similar:

“AI’s in rightward clusters are inversed, such that positive correlations denote positive asymmetry-asymmetry (genetic) relationships, regardless of direction of mean asymmetry in the cluster (i.e. higher asymmetry in the population-direction)”

**Author response image 1. sa2fig1:** Example of results in UKB using proposed alternative approach.

Heritability11. The account of results in the Abstract again focused on the positive findings and ignored the null results as if they were of no interest.So the Abstract states: "Areal asymmetry is moderately heritable (max h2SNP ~19%), and phenotypic correlations are reflected by high genetic correlations, whereas heritability of thickness asymmetry is low. "My reading of the results is that none of the twin-based heritability estimates – either global or regional – was statistically significant. Consistent with previous reports by Eyler et al. and others, the strongest influence on asymmetry indices is the E-term (nonshared environment), which may be interpreted in this context as chance (stochastic) influences.The SNP-based estimates from UK Biobank give a handful of significant heritability estimates for the SA asymmetry, but none for CT. I agree it is of interest to document these, but the genetic effects that are found are not high (19% is the highest estimate, with the remaining 80% of variance accounted for by the e2 term).

Following the Reviewers’ suggestion of switching to an AE model, we now have several FDR-corrected results also for the twin models. This makes the replication much clearer, at least for areal asymmetry. We agree that not enough attention was previously paid to the fact the E term soaks up most variance, and the potential theoretical interpretations of this. Please see our full response in the below comment (point 12). As mentioned above, the abstract now highlights that areal asymmetry was also low-moderately heritable, and invokes an interpretation of stochastic influences around these still relatively low estimates. However, we maintain our decision to highlight the 19% estimate, because to our knowledge this represents the most heritable brain or behavioural asymmetry yet reported with genomic methods, and this estimate is as high as many other structural brain imaging phenotypes that are not asymmetry-focused (Elliott et al. 2018). The fact that this same cluster associates with handedness will thus be of great interest to asymmetry researchers.

38 – “… Areal asymmetry correlates phenotypically and genetically in specific regions, and is low-moderately heritable (max h^2^_SNP_ ~19%). In contrast, thickness asymmetry is globally interrelated …, and exhibits low or absent heritability. … Results suggest areal asymmetry is developmentally stable and arises in early life through genetic but mainly subject-specific stochastic effects, whereas childhood developmental growth shapes thickness asymmetry …”

12 I guess it is a case of one's perspective, but my take on this is not "isn't it interesting that if we have a huge sample we can pick up some small genetic effects on a handful of asymmetry measures", but rather, "isn't it interesting that brains end up with similar patterns of structural asymmetry, even though genes explain so little individual variation ". This does suggest that there are genetic influences that create these general biases to one direction or the other, but which have reached fixation in the population and so show little individual variation.

Thank you. We fully agree this is a likely interpretation, and now also emphasize this in the Discussion (in addition to the Abstract):

435 – “This evident consensus suggests genetic-developmental programs regulate mean brain lateralization with respect to both area and thickness in humans. However, the genetic findings presented herein suggest these may have reached population fixation, as heritability of even our optimally delineated asymmetry measures was generally low. This indicates either subject-specific stochastic mechanisms in early neurodevelopment or later developmental influences primarily determine cortical asymmetry. Tracing their lifespan development, we show the lifespan trajectories of areal asymmetry primarily suggest this form of cerebral asymmetry is developmentally stable at least from age ~4, maintained throughout life, and formed early on – possibly in utero (Li et al. 2014; Sha, Schijven, et al. 2021). One interpretation of lifespan stability combined with low heritability may be stochastic early life developmental influences determine interindividual differences in areal asymmetry more than later developmental change. However, future work linking prenatal and childhood trajectories is needed to affirm this. Still, we also found relatively stronger heritability and reproducible heritability maps for areal asymmetry (notably, anterior insula exhibited ~19% SNP-based heritability). This also illustrates region-dependent interindividual genetic effects upon areal asymmetry, and the fact that phenotypic correlations were underpinned by high genetic correlations suggests specific areal asymmetries are formed under common genetic influence. In stark contrast, our findings of childhood development of thickness asymmetry until a peak around age ~24, higher directional variability in adult samples, and lower heritability, converge to suggest thickness asymmetry may be more shaped through subject-specific effects in later childhood development, possibly as the brain grows in interaction with the environment. This interpretation applied to asymmetry agrees with work suggesting cortical area in general may trace more to early life factors (Walhovd et al. 2016; Grasby et al. 2020) whereas thickness may be more related to and impacted by lifespan influences (Rakic 1995; Walhovd et al. 2016; Grasby et al. 2020)”

549 – “However, as noted above, although heritability of either hemisphere was high, areal asymmetry heritability was still only moderate at best, suggesting both genetics but primarily subject-specific stochastic effects likely underly its formation.”

Associations with cognition13. As noted above, I felt this treatment was too brief. There is a rich literature on associations with asymmetry for the variables considered here, much of it focusing on the degree rather than the direction of asymmetry. So I would like to see a separate paper dealing with associations with AIs from this analysis, relating the findings more systematically to prior studies.

We agree the literature review regarding asymmetry-cognition associations lacked some necessary detail. However, although a rich literature has attempted to associate diverse aspects of brain asymmetry with cognitive ability, almost none of these studies focuses specifically on cortical asymmetry. Indeed, we are only aware of three other studies testing cognitive links with cortical asymmetry or derivations of it (beyond our previous one which included cognitive change analysis in aging; (Roe et al. 2021)). Two of these focused on fluctuating asymmetry (Yeo et al. 2016; Moodie et al. 2020), operationalizing this as the average across absolute asymmetry deviance scores from the mean per-ROI, and are therefore not directly comparable to the present study. The remaining one found that greater thickness asymmetry in the expected direction may relate to better cognitive outcomes (Plessen et al. 2014). We now explicitly mention this study in the Introduction and Discussion, as well as summarize all previous work attempting to relate cortical asymmetry specifically to variations in cognitive ability:

106 – “Finally, altered development of cerebral lateralization in general has been widely hypothesized to relate to average poorer cognitive outcomes (Crow et al. 1998; Hirnstein et al. 2010; Plessen et al. 2014). Specifically in the context of cortical asymmetry, however, although one previous study reported larger thickness asymmetry may relate to better verbal and visuospatial cognition (Plessen et al. 2014), phenotypic asymmetry-cognition associations have been rarely reported (Plessen et al. 2014; Yeo et al. 2016; Moodie et al. 2020), conflicting (Yeo et al. 2016; Moodie et al. 2020), not directly comparable (Plessen et al. 2014; Moodie et al. 2020), and to date remain untested in large-scale data. Still, recent work points to small but significant overlap between genes underlying multivariate brain asymmetries and those influencing educational attainment and specific developmental disorders impacting cognition (Sha, Schijven, et al. 2021), indicating either pleiotropy between non-related traits or capturing shared genetic susceptibility to altered brain lateralization and cognitive outcomes.”

574 – “Also of note, no other asymmetry showed evidence of cognitive association (Figure 7—figure supplement 1), suggesting previously reported associations in small samples will likely not replicate (Plessen et al. 2014)”

Since we are careful not to generalize our findings to asymmetry measures not studied here, we do not agree we should separate this into another paper. Methodologically, we maintain our decision to reduce the 11 UK Biobank cognitive variables to their shared variation (Lyall et al. 2016), because the brief nature of UKB cognitive tests does not permit an overly thorough investigation into how different aspects of cognition relate to cortical asymmetry. That is, although it seemed like a brief treatment, it is a defensible and possibly close to optimal approach to take with this data. Note that we now consistently refer to our PC1 measure in terms of “general cognitive ability”. Regarding the distinction with absolute asymmetry, the statements we opt for are always specific to cortical asymmetry, and we are not aware of a large literature looking at absolute strength of cortical asymmetries when testing associations with cognition. However, we have added text to acknowledge this remains to be tested:

657 – “Relatedly, because UKB cognitive data is not exhaustive (Fawns-Ritchie and Deary 2020) (e.g. fluid IQ ranges from 1-13), we extracted the common variance across core tests to index general cognitive ability. This approach does not permit testing associations with well-operationalized or specific cognitive domains (for which UKB cognitive data may not be sufficient), and it remains to be seen whether cortical asymmetry may be informative in the context of specific forms of lateralized cognition (Ocklenburg et al. 2014), or using absolute non-directional measures. However, the subtlety of the only effect on cognition we find here may suggest we might expect similarly weak effects, though not necessarily if brain and phenotypic measurement accuracy is jointly optimized (Aki Nikolaidis et al. 2022).”

14. The discussion did a phenomenal job in pulling everything together but would benefit from some subheadings.

Thank you. We have added subheadings.

[Editors’ note: further revisions were suggested prior to acceptance, as described below.]

Essential revisions (for the authors):1) Thorough editions for clarity. This new reviewer has overall praise for your work, but they also expose the remaining lack of clarity and a relative lack of depth in coverage of the genetic results. Please use the detailed comments below as a guide when revising your study prior to final acceptance.

We thank the editor for their valuable input. We have made many editions for clarity as pointed out by the Reviewer. We have also substantially streamlined and shortened the Discussion.

2) Reproducibility: In addition to revising along the reviewer's comments, …– Please also fix issues such as "All maps are available at neurovault.org/XXXX."­

All maps have been made available in Supplementary file 2 and the text links have been updated.

­– Please add version numbers where applicable (eg to R packages).

This issue has been fixed.